# Grouter: Decoupling Routing from Representation for Accelerated MoE Training

**Yuqi Xu** [† 1 2]  **Rizhen Hu** [1]  **Zihan Liu** [3]  **Mou Sun** [2]  **Kun Yuan** [4]

## Abstract

Traditional Mixture-of-Experts (MoE) training typically proceeds without any structural priors, effectively requiring the model to simultaneously train expert weights while searching for an optimal routing policy within a vast combinatorial space. This entanglement often leads to sluggish convergence and training instabilities. This paper introduces GROUTER, a preemptive routing method that by distilling high-quality structures from fully-trained MoE models and serving as a fixed router for target models. By decoupling structural optimization from weight updates, GROUTER significantly accelerates both the speed and quality of model convergence. To ensure the framework's versatility, we also introduce expert folding to adapt GROUTER across varying model configurations and expert tuning to rebalance workloads across different data distributions. Furthermore, by leveraging the structural priors provided by preemptive routing, we can implement targeted optimizations to further enhance training throughput. Experiments demonstrate that GROUTER achieves superior performance and efficiency which boosts pre-training data utilization by **4.28**× and achieves up to **33.5**% throughput acceleration, establishing preemptive routing as a fundamental paradigm for scalable MoE training. We publicly release our code and pretrained GROUTER checkpoints [1].

## 1. Introduction

Large Language Models have rapidly become a foundation of modern AI, exhibiting strong capabilities in understand-

†: The research was finished while the author was an intern at Zhejang Lab. [1]School of Mathematical Sciences, Peking University, Beijing, China [2]Zhejiang Lab, Hangzhou, China [3]Yuanpei College, Peking University, Beijing, China [4]Center for Machine Learning Research, Peking University, Beijing, China. Correspondence to: Kun Yuan <kunyuan@pku.edu.cn>, Mou Sun <123sssmmm@gmail.com>.

*Proceedings of the 43rd International Conference on Machine Learning*, Seoul, South Korea. PMLR 306, 2026. Copyright 2026 by the author(s).

ing and reasoning (Bahdanau et al., 2014; Vaswani et al., 2017; Liu et al., 2019; Brown et al., 2020; Liu et al., 2024). A central driver of these gains is scaling: larger models, trained on more data, tend to deliver better performance (Kaplan et al., 2020). Yet naively scaling dense Transformers to the trillion-parameter regime is prohibitively expensive. Mixture-of-Experts (Shazeer et al., 2017) architectures address this tension by replacing dense feed-forward layers with a pool of experts and a router that activates only a small subset per token. This sparse activation allows parameter counts to grow without a proportional increase in per-step FLOPs, enabling training and inference at practical cost (Du et al., 2022; Wei et al., 2026).

Despite their theoretical efficiency, MoE models are notoriously difficult to train (Lepikhin et al., 2020). A fundamental cause of this difficulty is the tight coupling between routing structure learning and representation learning. In standard MoE training, the router and the experts are optimized simultaneously. The router must learn to partition the input space into balanced expert assignments, while the experts must simultaneously adapt their parameters to the evolving token distributions assigned to them. As illustrated in Figure 1a, our empirical observations reveal that the routing structure remains highly unstable even after extensive training, with expert assignments for identical inputs fluctuating significantly. This concurrent optimization of the router and representations forces experts to chase a "moving target" of shifting data distributions, preventing them from achieving deep specialization. Consequently, this joint training process—rather than the MoE architecture itself—leads to insufficient convergence and poor learning efficiency. We further formalize this destructive interference within a mathematical framework in Appendix A, where a gradient-based analysis elucidates how such concurrent updates hinder the development of specialized representations.

Several recent efforts have attempted to refine this routing process. Methods such as Auxiliary Loss Free (Wang et al., 2024a), differentiable routers like ReMoE (Wang et al., 2024b) or Lory (Zhong et al., 2024) aim to optimize the router more effectively, while TC-MoE (Yan et al., 2025), MoE++ (Jin et al., 2024) and Elastic MoE (Gu et al.,

---

[1]https://github.com/JimmyAwoe/Grouter

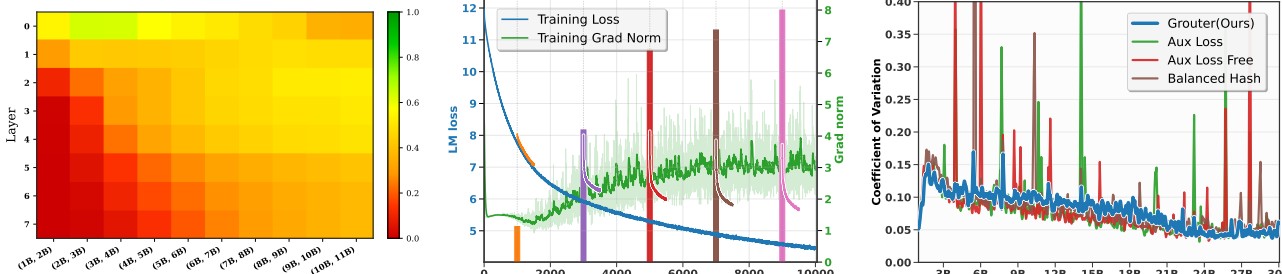

*Figure 1.* (a) The percentage of tokens that maintain an exactly identical set of **k** activated experts for the same input across adjacent checkpoints. We present the more detailed descriptions and figures in Appendix B. (b) Sensitivity analysis of expert specialization via random routing perturbations. Perturbations were applied at specific intervals with a fixed learning rate of $10^{-5}$. The resulting loss trajectories (lines) and average gradient norms (bars) reveal a clear trend: while early-stage training (1k steps) is resilient to routing noise due to a lack of specialization, later stages exhibit severe instability and loss spikes under perturbation. This demonstrates that experts gradually develop deep specialization as the router stabilizes, making the model increasingly sensitive to routing errors. (c) The coefficient of variation of the gradient norm within a sliding window of length 100. The results demonstrate that the Grouter method achieves a significant improvement in stability, whereas both the state-of-the-art load balancing approach and pure static routing assignment lead to substantial gradient fluctuations.

2026b), introduce dynamic expert selection to increase structural flexibility. However, these approaches still require the model to construct its load-balanced structure on-the-fly during training. By performing structural search and representation learning within the same optimization loop, they fail to resolve the underlying instability: the router is forced to partition a feature space that is itself shifting, while experts must specialize on tokens that they may not consistently receive in subsequent iterations. While some preliminary attempts have been made to decouple the router from the MoE training process, significant challenges remain; we provide a detailed discussion on the limitations of these specific approaches in Appendix C.

To overcome these challenges, this paper introduces the **General Router**, abbreviated as GROUTER. It learns a high-quality, well-optimized structure by distilling it from a fully trained MoE model (e.g. Qwen3-30B-A3B). GROUTER avoids the pitfalls of manual design and circumvents the instability of learning from a dynamically evolving model by extracting structure from a stable, converged state. Once distilled, GROUTER serves as a plug-and-play replacement for the trainable router and offers three key advantages. **First**, GROUTER inherits high-quality structural priors from converged models, surpassing those learned from scratch. **Second**, by fixing the routing pathways, it decouples routing topology from representation, thereby eliminating optimization interference and focusing training resources on task performance. **Third**, this decoupled structure serves as a stable foundation that facilitates the injection of further routing-based optimizations.

GROUTER distills structural knowledge from a fully trained MoE model. To ensure versatility, we incorporate two key mechanisms: Expert Tuning, which rebalances workloads to adapt to specific data distributions, and Expert Folding, which enables the router to fit varying MoE configurations.

This design allows a single distilled GROUTER instance to be efficiently transferred across diverse training scenarios.

We summarize our key contributions as follows:

1. **Analyzing the Necessity of Decoupling in MoE Training.** We empirically demonstrate that the entanglement between routing structure and representation learning restricts MoE scaling. We show that decoupling these processes is critical for achieving optimal convergence speed and training stability.

2. **Introducing Grouter for Preemptive Structure Construction.** We propose GROUTER, which distills optimal routing structures from converged models. By establishing a fixed routing topology prior to training, GROUTER fundamentally eliminates the interference between structure learning and representation updates.

3. **Expanding the Optimization Space via Structural Priors.** Leveraging the fixed priors provided by GROUTER, we shift data optimization from runtime to a pre-processing stage. This decoupling bypasses the limitations of dynamic routing, enabling the application of sophisticated offline algorithms to significantly expand the optimization space.

We implemented GROUTER within the Megatron-LM (Shoeybi et al., 2019) and conducted comprehensive experiments utilizing clusters of NVIDIA H100 and A100. Our pre-training results on a 550M parameter MoE model demonstrate remarkable data efficiency: GROUTER achieved the same validation set loss as the baseline model using only **23.3%** of the data, corresponding to a **4.28×** improvement in data utilization efficiency. Furthermore, GROUTER consistently exhibited significant advantages across experiments with different architectures (OpenAI, 2025; DeepSeek-AI, 2024; Team, 2025) and scales. We also

performed throughput experiments, showing that GROUTER can achieve up to a **33.5%** throughput increase under appropriate Expert Parallelism settings. Collectively, these results demonstrate the comprehensive benefits and advancements that GROUTER brings to MoE. We will open-source our code and GROUTER checkpoints to facilitate community adoption and further improvements.

## 2. Background and Motivation

### 2.1. Mixture-of-Experts

The MoE architecture has emerged as a critical paradigm for scaling Transformer models (Shazeer et al., 2017). Its primary distinction from dense Transformers lies in the Feed-Forward Network (FFN) layer, which in MoE comprises a router and a set of $E$ experts. During the forward pass, the router assigns $k$ experts to each input token, achieving sparse parameter activation. The FFN component of a single MoE layer can be formally described as:

$$\mathbf{y} = \sum_{i=1}^{E} \mathbf{r}(\mathbf{x})_i \mathbf{f}_i(\mathbf{x}), \tag{1}$$

$$\mathbf{r}(\mathbf{x}) = \mathbf{g}\left(\text{TopK}(\mathbf{s}(\mathbf{x}), k)\right). \tag{2}$$

Here, $\mathbf{x} \in \mathbb{R}^d$ denotes the input to the MoE layer, and $\mathbf{s}(\mathbf{x}) \in \mathbb{R}^E$ represents the raw scores assigned by the router to the $E$ experts. The $\text{TopK}(\cdot, k)$ operator retains only the scores of the $k$ highest-scoring experts. Subsequently, $\mathbf{g}(\cdot)$ is a normalization function, typically Softmax or Sigmoid, applied to these selected scores to yield the final coefficient vector $\mathbf{r}(\mathbf{x}) \in \mathbb{R}^E$. Specifically, $\mathbf{r}(\mathbf{x})_i$ denotes the weight applied to the output of expert $\mathbf{f}_i(\mathbf{x})$. The MoE output $\mathbf{y}$ is then passed through subsequent layers, eventually contributing to the final task loss $\mathcal{L}$ (e.g., cross-entropy for language modeling). It is noteworthy that although the notation here only explicitly shows the Router network, the overall structure is implicitly influenced by all preceding network parameters.

### 2.2. Inefficiency of Joint Router-Expert Optimization

We conducted an empirical study on the expert specialization (Hu et al., 2026a) levels across different training stages. Specifically, we introduced stochastic perturbations by randomizing the router's outputs at various intervals and analyzed the subsequent fluctuations. A sharp spike in gradient norms and loss suggests that the experts are unable to adapt to arbitrary token assignments. Consequently, this heightened sensitivity to routing noise reflects a deeper degree of expert specialization.

As illustrated in Figure 1b, a clear trend emerges: in the early stages of training, the experts exhibit a relatively low degree of specialization; thus, abrupt randomization of the routing logic does not trigger training collapse, as the experts have

not yet formed distinct niche representations. However, as training progresses, the level of specialization significantly deepens. In conjunction with Figure 1a, we identify routing instability as the primary cause of this early-stage behavior: the fluctuating routing policy subjects experts to constantly shifting objectives, preventing them from specializing. Once the routing stabilizes, experts are able to commit to specific domains, leading to the observed increase in specialization.

We further investigate the coefficient of variation (CV) of the gradient norm throughout the training process, as shown in Figure 1c. We observe that fluctuations in the gradient norm gradually emerge during the mid-to-late stages. We ascribe this phenomenon to the progressive deepening of expert specialization: as experts become highly specialized, occasional routing errors expose them to incompatible tokens, thereby triggering spikes in gradient magnitude.

These findings suggest that the learning process can be optimized by introducing a structural prior from the beginning, which provides a consistent data distribution for each expert. This decoupling allows the model to bypass the volatile exploration phase and focus directly on expert specialization during the subsequent training process, thereby significantly enhancing overall training efficiency.

We provide a more comprehensive discussion on other advantages of this decoupling strategy in Appendix D.

## 3. Method

The core objective of GROUTER is to fundamentally decouple the routing structure from representation learning by proactively extracting a stable, high-quality routing structure $\mathbf{r}^*(\cdot)$ from a well-optimized source MoE model. This extraction process involves GROUTER learning to replicate the input-to-expert mappings and weight assignments produced by the source model's router. The resulting fixed structure, provided by the frozen GROUTER network, then serves as a preemptive structural prior for target MoE model training. This approach fundamentally decouples structural optimization from task performance optimization, ensuring efficient training.

### 3.1. Grouter Architecture and Structure Extraction

#### 3.1.1. GROUTER ARCHITECTURE

The GROUTER network, denoted as $\mathbf{G}$, is designed as a lightweight, standalone structure extractor. Its primary role is to learn the desired routing decisions $\mathbf{r}^*(\cdot)$ from the source model. Unlike traditional MoE routers that operate within the model backbone, GROUTER processes raw token sequences $\mathbf{X}$ directly from the tokenizer, operating entirely independently. The architecture comprises an embedding layer, $N$ Transformer encoder blocks, and a final linear

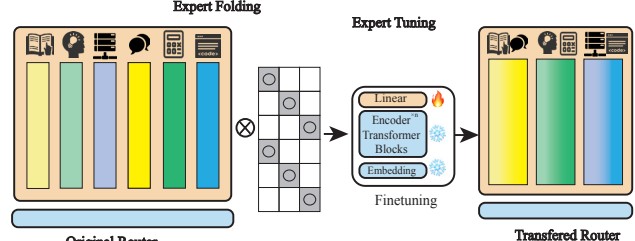

*Figure 2.* (a) Overview of the GROUTER Workflow. The GROUTER first extracts a highly optimized structural prior from the Source Model, and then injects this prior into the Target Model in a frozen state. (b) Illustration of Our Expert Tuning and Expert Folding Techniques.

projection layer. Formally, **G** can be expressed as:

$$\mathbf{G}(\mathbf{X}) = \mathbf{W}_s \left( \text{Enc}^N \left( \text{Emb}(\mathbf{X}) \right) \right). \quad (3)$$

Here, $\text{Emb}(\cdot)$ denotes the embedding layer, $\text{Enc}^N$ represents a stack of $N$ Transformer encoder blocks, and $\mathbf{W}_s$ is the final linear projection layer. The use of Transformer encoder blocks enables **G** to capture global context, allowing it to assess the contextual relevance of inputs when determining optimal expert assignments. This lightweight architecture minimizes computational overhead during pre-training. We present experiments in Appendix G that demonstrate the advantages of this architecture.

### 3.1.2. STRUCTURE EXTRACTION

As seen in Figure 2a, We employ knowledge distillation to extract the high-quality stable routing structure from a large, fully converged source MoE model into the lightweight GROUTER network. The distillation objective is to train GROUTER to precisely replicate the expert assignment weights of the source model's router, thereby obtaining the desired preemptive router $\mathbf{r}^*(\cdot)$.

For a given input $\mathbf{X}$, let $\mathbf{H}$ be the hidden state that $\mathbf{X}$ propagates to at the input of the selected router layer in the source model $\mathbf{S}$. The raw expert scores output by this router are denoted by $\mathbf{s}_{\text{Sou}}(\mathbf{H})$. The distillation loss $\mathcal{L}_{\text{Distill}}$ is formulated using the Kullback-Leibler divergence:

$$\mathcal{L}_{\text{Distill}} = D_{\text{KL}} \left( \text{Softmax}(\mathbf{s}_{\text{Sou}}(\mathbf{H})) \parallel \text{Softmax}(\mathbf{G}(\mathbf{X})) \right).$$

Unlike typical knowledge distillation, we do not incorporate a temperature parameter in the Softmax calculation. This is crucial because GROUTER must accurately learn the true magnitudes of expert contribution weights, not merely the rank ordering or relative differences of the raw logits.

**Shared GROUTER Implementation.** In multi-layer MoE architectures, we implement a single GROUTER to guide all MoE layers. This design is supported by empirically observed redundancy in layer-wise routing. Specifically, Cai et al. (2024) suggest that complex, layer-specific routing decisions are unnecessary, as high correlation exists among

the routing structures of different MoE layers. This inherent redundancy indicates that a single, universally applicable structural prior suffices for the entire network. Our subsequent experiments confirm that this shared GROUTER does not compromise model performance. This design choice is critical for maintaining the computational efficiency of our method.

**Layer Selection Strategy.** To maximize the quality of the structural prior derived from a single GROUTER, we distill the routing output from the first source MoE layer. This selection is motivated by the sequential nature of Transformers: routing deviations originating in early MoE layers are progressively amplified in deeper layers. This accumulation results in significantly higher routing fluctuation and lower structural stability in downstream layers. By distilling from the most stable routing pattern, GROUTER acquires a robust structural prior $\mathbf{r}^*(\cdot)$ that is minimally affected by cumulative errors.

### 3.2. Expert Folding and Expert Expanding

A critical challenge in adopting a fixed structural prior is its transferability across MoE models with varying expert configurations. Specifically, the number of experts in the source model from which GROUTER is distilled may differ from the number in the target model that GROUTER is intended to guide. To address this limitation, we propose Expert Folding and Expert Expanding, which enables a single GROUTER instance to flexibly adapt to varying expert counts.

### 3.2.1. EXPERT FOLDING

**Expert Folding Mapping.** The Expert Folding procedure leverages an affinity-based merging strategy to construct a mapping that reduces the source expert count $E_S$ to the target count $E_T$. The process begins by characterizing the functional relationships among source experts. We execute GROUTER on the training dataset to compute the Expert Co-activation Affinity Matrix $\mathbf{P} \in \mathbb{R}^{E_S \times E_S}$, where element $\mathbf{P}_{ij}$ quantifies how often source experts $i$ and $j$ are

simultaneously activated by GROUTER for the same input token. Next, we determine the required merging size for each of the $E_T$ target expert. Target experts are designed to incorporate $G_{\text{avg}} = \lfloor E_S/E_T \rfloor$ source experts on average, with $N_{\text{extra}} = E_S \bmod E_T$ groups requiring an additional expert. With group sizes established, we iteratively select an unassigned source expert $e_i$ to initiate a new merging group $C_k$. To maximize the resulting composite expert's effectiveness, we then repeatedly select the unassigned expert $e_j$ that maximizes collective co-activation affinity with the current group members:

$$\arg\max_{e_j \notin \text{Assigned}} \sum_{e_m \in C_k} \mathbf{P}_{e_m, e_j}. \tag{4}$$

This process continues until group $C_k$ reaches its target size. By mapping each $C_k$ to a target expert $e_k$, we ensure maximal preservation of essential structures and specialized function within the resulting composite target experts.

**Matrix Folding Implementation.** The folding procedure is implemented efficiently as a linear transformation applied to the final score layer of the distilled GROUTER as shown in Figure 2b. Let $\mathbf{W}_s \in \mathbb{R}^{d \times E_S}$ be the original weight matrix of GROUTER's final score layer, and let $\mathbf{M} \in \{0,1\}^{E_S \times E_T}$ be the binary mapping matrix derived from the affinity-based merging strategy, where $\mathbf{M}_{ij} = 1$ indicates that source expert $i$ is mapped to target expert $j$. The folded weight matrix for the target model, $\tilde{\mathbf{W}}_s \in \mathbb{R}^{d \times E_T}$, is computed as:

$$\tilde{\mathbf{W}}_s = \mathbf{W}_s \mathbf{M}. \tag{5}$$

This operation is computationally negligible and requires minimal storage. Because $\mathbf{M}$ can be precomputed and stored for various target configurations, a single distilled GROUTER instance can serve as a structural prior for diverse MoE models simply by selecting the appropriate $\mathbf{M}$. This enables exceptional transferability and configuration flexibility for GROUTER with minimal overhead.

### 3.2.2. EXPERT EXPANDING

**Underserved Tokens Identification.** The expansion procedure begins by identifying tokens for which the existing structural prior provides inadequate coverage. Specifically, we execute the frozen GROUTER on a representative corpus $\mathcal{D}$ and extract the hidden representations $\mathbf{h} \in \mathbb{R}^d$ at the input of the final score layer $\mathbf{W}_s$ via a forward hook. For each token, we compute the maximum expert affinity score $s_{\text{max}} = \max_j(\mathbf{h}^\top \mathbf{w}_j)$, where $\mathbf{w}_j$ denotes the $j$-th row of $\mathbf{W}_s$. Tokens whose $s_{\text{max}}$ falls below the $q$-th quantile of the score distribution are designated as *underserved tokens*, forming the set $\mathcal{U}$. Intuitively, these are tokens for which no existing expert direction provides a strong routing signal, and they constitute the natural demand for new experts.

**Centroid Clustering.** Given $E_\Delta = E_T - E_S$ new experts to

be constructed, we apply $k$-means clustering on the hidden representations of the underserved token set $\mathcal{U}$ with $k = E_\Delta$. The resulting centroids $\{\mathbf{c}_1, \ldots, \mathbf{c}_{E_\Delta}\} \subset \mathbb{R}^d$ identify $E_\Delta$ representative directions in the hidden-state space that characterize the distribution of tokens poorly served by the current routing structure. Each centroid provides the initial direction for one new expert.

**Orthogonal Projection and Normalization.** To ensure the new expert directions are maximally independent from the existing ones, we project the centroids onto the null space of the current score weight matrix. Let $\mathbf{A} = \mathbf{W}_s^\top \in \mathbb{R}^{d \times E_S}$. The projection is computed as:

$$\mathbf{c}_i' = \mathbf{c}_i - \mathbf{A}(\mathbf{A}^\top \mathbf{A})^{-1} \mathbf{A}^\top \mathbf{c}_i, \quad i = 1, \ldots, E_\Delta. \tag{6}$$

This removes all components aligned with existing expert directions, preventing the new experts from duplicating learned routing signals. Subsequently, we apply QR decomposition (Golub & Van Loan, 2013) to the projected centroid matrix to enforce mutual orthogonality among the new experts themselves. Finally, each new direction is rescaled to match the average $\ell_2$-norm of the original expert weight vectors, ensuring consistent scoring magnitudes across old and new experts.

**Weight Assembly.** The expanded weight matrix is constructed by concatenating the original and new expert directions:

$$\mathbf{W}_s^{\text{exp}} = \begin{bmatrix} \mathbf{W}_s \\ \mathbf{W}_\Delta \end{bmatrix} \in \mathbb{R}^{E_T \times d}, \tag{7}$$

where $\mathbf{W}_\Delta \in \mathbb{R}^{E_\Delta \times d}$ contains the orthogonalized and normalized new expert directions. Critically, the original weights $\mathbf{W}_s$ remain unchanged, preserving the full structural prior learned during distillation.

### 3.3. Expert Tuning

Once the GROUTER network $\mathbf{G}$ is distilled and expert folding is applied, its weights are frozen, yielding the fixed structural prior $\mathbf{G}^*$. This $\mathbf{G}^*$ provides constant, preemptive routing decisions for the target MoE model, successfully decoupling structure from performance optimization.

However, the load balancing of $\mathbf{G}^*$ may be suboptimal. This suboptimality does not stem from failures in structure extraction; rather, it arises because the structural prior distilled from the source model is optimized for its training data distribution $\mathcal{D}_S$, which may differ from the target model's deployment distribution $\mathcal{D}_T$. Due to inherent expert specialization and task-optimal routing patterns learned on $\mathcal{D}_S$, the distilled $\mathbf{G}^*$ naturally inherits a routing bias that, when applied to $\mathcal{D}_T$, leads to imbalanced load distribution.

To mitigate this imbalance without compromising the stable structure, we perform a lightweight fine-tuning of the GROUTER based on the Target Model's training distribution

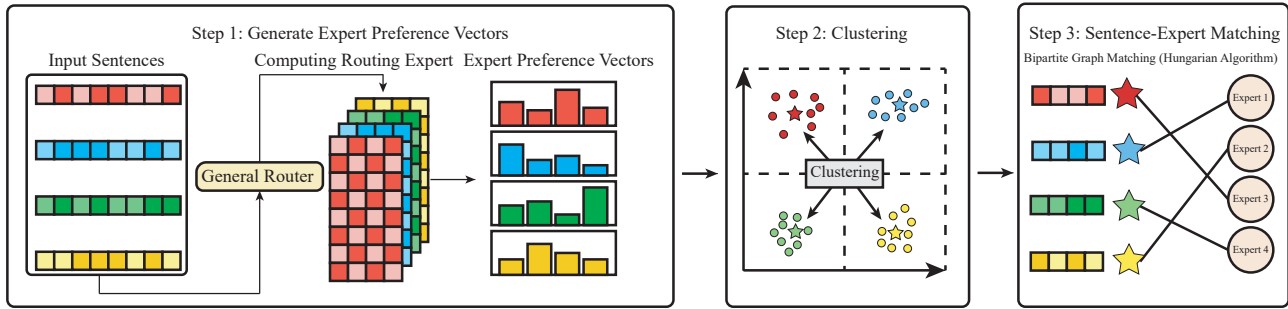

*Figure 3.* The workflow of our EP communication optimization strategy. Step 1: Sequences pass through GROUTER to generate token-level assignments, which are aggregated into sequence-level routing affinity vectors. Step 2: Sequences are clustered based on affinity. Using cluster centroids as preference weights, we solve an optimization problem to assign experts to EP devices, maximizing the alignment between experts and sequence clusters. Step 3: With expert locations fixed, sequences are assigned to the EP device that minimizes the resulting communication volume.

$\mathcal{D}_T$ before the main training phase as seen in Figure 2b. We adopt the $\mathcal{L}_{\text{aux}}$ load balancing loss, similar to that used in (Lepikhin et al., 2020), as the sole optimization objective to encourage the GROUTER to adapt to the new distribution.

Crucially, to maximally preserve the structure learned during distillation, we freeze all GROUTER parameters except for the final linear projection layer. This fine-tuning is applied only to this negligibly small parameter count. Our subsequent empirical results demonstrate that this adjustment requires only a minimal amount of data to achieve a perfectly usable level of load balance. This feature allows the GROUTER to be conveniently transferred with minimal overhead to target training tasks.

### 3.4. Enhancing Training Efficiency via Preemptive Routing

The fixed and decoupled structural prior $\mathbf{r}^*(\mathbf{X})$ provided by the frozen GROUTER effectively decouple the routing structure from representation learning. This stable routing decision simultaneously presents a unique opportunity to significantly enhance training efficiency. Unlike traditional methods (He et al., 2022; Nie et al., 2023; Liu et al., 2025; Zhang et al., 2025) that must operate on dynamic, real-time routing decisions, GROUTER offers a preemptive, known routing map throughout the entire training process, enabling offline optimization and eliminating runtime overhead.

#### 3.4.1. DECOUPLING PREEMPTIVE ROUTING

Given that the GROUTER is completely decoupled and fixed, we shift the routing task from the model's forward pass to the data preprocessing pipeline. Specifically, the input data is processed by the GROUTER network ahead of time, and the resulting routing decisions are then cached and stored as part of the processed dataset. During the model's forward pass, these stored decisions are loaded and passed directly to guide the MoE layers.

This mechanism offers a trade-off between storage and computation. For each token, only the active expert indices and their corresponding gating weights need to be stored. Critically, since the routing is computed only once but can be reused across multiple training epochs, the GROUTER provides an efficient pathway to leverage storage capacity for runtime acceleration.

#### 3.4.2. DECOUPLING COMMUNICATION OPTIMIZATION

Training large MoE models often relies on Expert Parallelism (EP), which distributes experts across devices. This requires all-to-all communication during the token dispatch phase, where the dynamic nature of routing necessitates synchronous communication optimization.

Another key innovation of the GROUTER framework is to decouple this communication optimization from the synchronous training path as illustrated in Figure 3. We define the routing affinity vector $\phi(\mathbf{X}) \in \mathbb{R}^{\mathbf{E}_T}$ of a sample $\mathbf{X}$ as the average expert selection frequency across all tokens $t \in \mathbf{X}$:

$$\phi(\mathbf{X})_i = \frac{1}{|\mathbf{X}|} \sum_{t \in \mathbf{X}} \mathbb{I}\left[e_i \in \text{TopK}(\mathbf{G}^*(\mathbf{x}_t), k)\right] \quad (8)$$

where $\mathbb{I}[\cdot]$ is the indicator function, $\mathbf{e}_i$ is expert $i$ and $\mathbf{G}^*$ is the frozen GROUTER.

Leveraging $\mathbf{G}^*$, we collect the routing affinity vectors for all training samples to construct the full affinity set

$$\Phi = \{\phi(\mathbf{X}_i) \mid \mathbf{X}_i \in \mathcal{D}_{\text{train}}\} \quad (9)$$

This set serves as the basis for the subsequent structural optimization.

**Expert Grouping.** The first step for optimization is to establish fixed expert groups that are co-located on physical devices. Each sample's affinity vector $\phi(\mathbf{X})$ indicates its overall preference for the $E_T$ experts. We apply a clustering algorithm to the set of affinity vectors $\Phi$. The number of

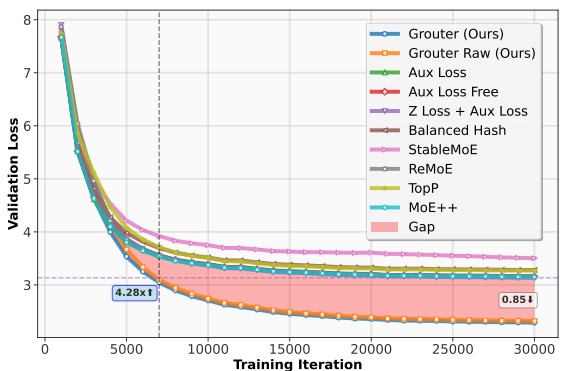 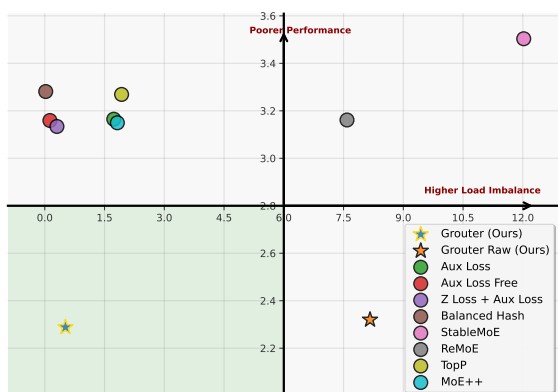

*Figure 4.* (a) Pre-training Validation Loss Curves Across 30B Tokens. Grouter Raw denotes the distilled Grouter used without the subsequent Expert-Tuning. The shaded Gap region illustrates the loss difference between the Grouter curve and the best baseline. Overall, Grouter achieves a $4.28\times$ acceleration or a maximum loss reduction of $0.85$ at the same training data volume. (b) Comparison of Performance and Load Balance Trade-off. The third quadrant is highlighted in green to emphasize that models located within this region achieve an optimal balance between low load violation and superior model performance

clusters is explicitly set to $\mathbf{N}_p$, which is the total number of Expert Parallel partitions. Subsequently, experts are assigned to these clusters with the objective of maximizing the aggregated affinity, thereby constructing $\mathbf{N}_p$ distinct expert groups:

$$\{\mathcal{E}_p\}_{p=1}^{N_p} = \text{Clustering}(\Phi, N_p) \quad \mathcal{E}_p \subset \{1, \ldots, E_T\} \tag{10}$$

where $\{1, \ldots, E_T\}$ is the experts of Target MoE Model. Each cluster $\mathcal{E}_p \subseteq \{1, \ldots, E_T\}$ represents an Expert Group, a subset of experts that are frequently co-activated by a specific type of input sample, which are subsequently mapped to the $\mathbf{N}_p$ physical EP partitions.

**Sample Placement Optimization.** Once the Expert Groups are fixed, we determine the optimal placement for each input sample $\mathbf{X}_i$ by minimizing the communication cost. Specifically, the optimal partition ID is determined by:

$$\text{PartitionID}(\mathbf{X}_i) = \underset{p \in \{1, \ldots, N_p\}}{\arg\min} \left(\text{Cost}(\mathbf{X}_i, \mathcal{E}_p)\right) \tag{11}$$

where $\text{Cost}(\mathbf{X}_i, \mathcal{E}_p)$ represents the communication cost incurred by assigning sample $\mathbf{X}_i$ to the partition containing Expert Group $\mathcal{E}_p$. This optimization yields $\mathbf{N}_p$ fixed Sample Groups, where each corresponds to one EP partition. During training, samples are statically assigned to their corresponding EP group based on $\text{PartitionID}(\mathbf{X}_i)$.

This optimization process fundamentally transforms the challenge of dynamic, runtime communication into a manageable, pre-computed resource allocation problem. By utilizing the fixed structural prior of GROUTER to define expert groups and sample groups, we effectively eliminate the high latency and overhead of synchronous communication optimization, leading to significant acceleration and stability in MoE training.

## 4. Experiment

### 4.1. Experiment Setup

**Infrastructure** Our experiments were conducted on a cluster comprising eight NVIDIA A100 GPUs and eight NVIDIA H100 GPUs, interconnected via NVLink. The software environment utilized PyTorch 2.8.0 with CUDA 12.9 and the Megatron-LM framework (commit hash e7c55de9).

**Grouter Distillation Setup** We distill our GROUTER from the Qwen3-30B-A3B model (Team, 2025). The distillation process is performed on the C4 dataset(Raffel et al., 2020) for 2.6 Billion tokens. We construct the GROUTER using a three-layer Transformer Encoder, resulting in a total parameter count of 60M. Notably, 50M of these parameters reside in the computationally inexpensive Embedding layer, which makes the overall GROUTER architecture highly lightweight. The consistently decreasing loss curve, as shown in Appendix M, demonstrates the successful extraction of the structural prior. Unless otherwise specified, all subsequent experiments are based on this single distilled GROUTER instance.

**Model Configurations** We define five distinct MoE model variants for our experiments: Tiny-Qwen3, Mini-Qwen3, Samll-Qwen3, Mini-DS-V2-Lite, and Mini-GPT-OSS. These models are structurally based on the architectures of Qwen3-30B-A3B (Team, 2025), DeepSeek-V2-Lite (DeepSeek-AI, 2024), and GPT-OSS-20B (OpenAI, 2025), respectively. Their detailed configurations are presented in the Appendix L.

### 4.2. Performance

Model performance is measured using the validation set cross-entropy loss, with the results presented in Figure 4a.

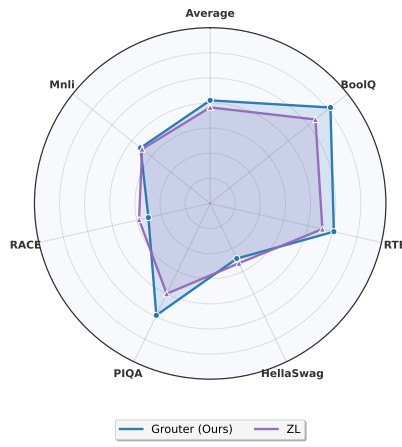 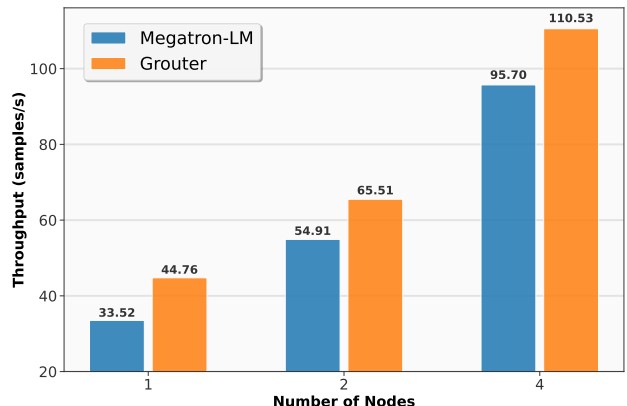

*Figure 5.* (a) Downstream Task Results: GROUTER achieves an average improvement of 2.80 across six benchmarks, with gains of up to 10 points on specific tasks. This demonstrates that the reduced validation loss achieved by GROUTER translates into genuine enhancements in model capabilities, rather than merely overfitting the validation metric. (b) Throughput Scaling: We evaluate throughput on 1, 2, and 4 nodes with Expert Parallelism (EP) degrees set to 8, 16, and 32, respectively. For the multi-node setups (2 and 4 nodes), we apply node-granularity communication optimization, while for the single-node case, we utilize GPU-granularity optimization.

Each model is trained for 30 Billion tokens, which is enough to achieve convergence according to the compute-optimal dataset size predicted by Krajewski et al. (2024). We distinguish between two GROUTER variants: **Grouter Raw**, which denotes the direct use of the distilled GROUTER without the subsequent load-balancing expert tuning, and **Grouter**, which refers to the version subjected to this fine-tuning on only 50M tokens of the C4 dataset. The implementation details and specific configurations of the other baselines are provided in Appendix F.

Our results demonstrate significant efficiency gains: GROUTER achieves the same convergence loss by leveraging only **23.3**% of the training data, which corresponds to a **4.28×** acceleration in convergence speed. Furthermore, when compared at the equivalent training volume, GROUTER exhibits superior final performance, achieving a loss reduction of up to $0.85$. This indicates that, rather than constraining the model capacity, the fixed structure of GROUTER actually facilitates superior convergence accuracy. We also observe that the performance gap consistently widens as training progresses. We attribute this sustained advantage to the fact that GROUTER completely eliminates the optimization interference arising from the synchronous learning of routing policies and representations. Finally, the marginal difference between **Grouter Raw** and the fine-tuned **Grouter** validates that our lightweight fine-tuning strategy successfully retains the high-quality structural information initially extracted from the teacher model.

Prior research suggests that excessive pursuit of load balancing can compromise model performance(Wang et al., 2024a). In this context, we show that Grouter achieves an excellent balance, significantly enhancing performance without unduly sacrificing load stability.

To quantify this, we measure the MaxVio$_{Global}$ metric defined in Appendix H for all MoE models after 30B training tokens. The validation set loss is used as the complementary measure of performance. **The results, displayed in Figure 4b, confirm that GROUTER maintains competitive load balance despite its superior performance gains over all compared methods.** Significantly, the **Grouter Raw** variant simultaneously exhibits severe load imbalance. This finding directly corroborates our discussion in Section 3.3, validating that the disparity in data distributions between the source model and the target model may cause significant load imbalance observed in the directly distilled GROUTER.

To demonstrate that GROUTER can be effectively transferred across different model configurations, we validated its scalability on four models varying in size, architecture, and expert setup. We first utilize expert folding or expert expanding to adapt the structural prior to target models with fewer or more experts than the source, respectively. Subsequently, the expert tuning technique is applied to ensure Grouter achieves adequate load balance across varying numbers of active experts. We compare the final validation set loss of our Grouter against the SOTA baseline from our pre-training experiments , Z-Loss + Aux Loss (abbreviated as ZL), after sufficient training. The results, presented in Table 1, clearly show that GROUTER demonstrates effective training acceleration across all tested scales and architectures. **This compelling outcome strongly validates the efficacy of our expert folding, expert expanding and expert tuning adaptation schemes.** The complete training curves for these experiments are provided in Appendix K.

We evaluated the Mini-Qwen3 model, pre-trained for 50B tokens using both GROUTER and the ZL baseline, on a suite of downstream tasks.The detailed experimental setup

*Table 1.* Performance Comparison Across Diverse Model Configurations

|  | **Mini-GPT-OSS** | **Mini-DS-V2-Lite** | **Mini-Qwen3** | **Small-Qwen3** |
|---|---|---|---|---|
| Attn | GQA | MLA | GQA | GQA |
| Structure | MoE | Dense+MoE | MoE | MoE |
| Tokens | 30B | 30B | 50B | 50B |
| Grouter | **2.940** | **2.469** | **1.763** | **1.648** |
| ZL | 3.346 | 3.057 | 2.740 | 2.633 |

is described in Appendix I, and the evaluation results are presented in Figure 5a.

The results show that Grouter consistently outperforms the ZL baseline across these evaluation tasks. **This finding strongly suggests that our method not only enhances training efficiency but also improves the model's overall problem-solving capability and generalization performance.**

### 4.3. Efficiency

Next, we leverage the prior knowledge acquired by GROUTER to enhance model training efficiency through router decoupling and data reorganization. Specifically, we first utilize GROUTER to obtain expert assignment information for the C4 dataset. Subsequently, we sequentially apply Expert Grouping and Sample Placement Optimization to maximize the reduction in EP communication overhead.

We conducted the communication saving experiments using the Mini-Qwen3 model. Crucially, all experiments were integrated with DeepEP (Liu et al., 2024), one of the most advanced EP communication saving systems. This integration demonstrates that our methodology can effectively combine with state-of-the-art EP communication techniques to further boost training efficiency.

We measure the throughput against a baseline using the standard Auxiliary Loss run on the Megatron-LM framework, specifically calculating the throughput averaged over the first 1000 training iterations. The experimental results, presented in Figure 5b, show that Grouter consistently achieves notable communication savings across various EP counts. Specifically, we observe throughput speedups of $33.5\%$, $19.3\%$, and $15.5\%$ on 1, 2, and 4 nodes, respectively. These improvements demonstrate that GROUTER significantly contributes to achieving higher training efficiency for MoE models. The detailed experimental settings is in Appendix J.

### 4.4. Discussion

We investigate the factors contributing to the superior performance of GROUTER. We quantify the volatility of the training process by measuring the gradient norms, as illustrated in Figure 1c. We observe that GROUTER exhibits an exceptionally stable coefficient of variation for the gradient norm, devoid of any spikes throughout the training

phase. In contrast, methods such as Aux Loss, Aux Loss Free and HashLayer suffer from frequent, high-magnitude fluctuations. We attribute this phenomenon to the routing instability in baseline methods, where experts occasionally encounter unfamiliar tokens. These tokens generate anomalous gradients that disrupt the expert's learned representations. Conversely, by establishing a structural prior for expert assignment, GROUTER facilitates continuous expert specialization on designated tasks, thereby significantly enhancing learning efficiency and ensuring training stability. A more detailed discussion is provided in Appendix N.

## 5. Conclusion

We have presented GROUTER, a preemptive routing framework that decouples routing structure optimization from representation learning in MoE training. By distilling high-quality routing decisions from a converged source model into a lightweight standalone network, GROUTER provides a fixed structural prior that eliminates the instability arising from joint router-expert optimization. Combined with expert folding, expert expanding, and expert tuning, a single distilled GROUTER instance adapts flexibly to diverse model scales and expert configurations. Extensive experiments demonstrate that GROUTER achieves a $4.28\times$ improvement in data utilization efficiency and up to $33.5\%$ throughput acceleration, while maintaining superior convergence quality across varying architectures. These results establish preemptive routing as a practical and effective paradigm for scalable MoE pre-training.

## 6. Future Works

GROUTER's ability to pre-acquire Dispatch information enables its integration with various communication saving and training acceleration methods. The deterministic routing provided by the frozen GROUTER also offers natural resilience to hardware failures, presenting a promising new avenue for algorithmic fault tolerance as explored by Hu et al. (2026b). Furthermore, this stability is crucial for Reinforcement Learning, where the expert-activation volatility of MoE models can prevent RL training from converging properly (Zheng et al., 2025; Ma et al., 2025). Since GROUTER is frozen during training, it bypasses this critical issue. We leave the detailed exploration of both these synergistic efficiency gains and post-training applications for future work.

## Acknowledgments

This work is funded by the National Key Research and Development Program of China (No. 2024YFA1012902) and the National Natural Science Foundation of China (No. 92370121, 12301392, 12288101, W2441021). This research is also supported by Zhejiang Lab and the AI for Science Institute, Beijing, China.

## Impact Statement

This research focuses on optimizing the training dynamics of sparse models. By decoupling routing from representation, our approach lowers the data and computational barriers for training high-performance MoE models, potentially democratizing access to large-scale AI capabilities. The proposed method is a general optimization technique and does not introduce specific potential for societal harm.

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

## A. Routing-Representation Interference

Suppose the MoE layer defined in (1)–(2) is parameterized by router parameters $\theta_r$ (within $\mathbf{s}(\mathbf{x})$) and expert parameters $\theta_{e_i}$ (within $\mathbf{f}_i(\mathbf{x})$). During training, these parameters are updated via gradient descent:

$$\theta_r^{t+1} = \theta_r^t - \alpha \nabla_{\theta_r}^t \mathcal{L}, \quad \theta_e^{t+1} = \theta_e^t - \alpha \nabla_{\theta_e}^t \mathcal{L} \qquad (12)$$

where $\mathcal{L}$ is the task loss and $e$ denotes all experts parameters. Due to the composite structure of the MoE output in (1), the gradients decompose into two components by chain rule:

$$\nabla_{\theta_r}^t \mathcal{L} = \frac{\partial \mathcal{L}}{\partial \mathbf{y}} \cdot \frac{\partial \mathbf{y}}{\partial \theta_r}\Big|_{\theta_r^t}, \quad \nabla_{\theta_e}^t \mathcal{L} = \frac{\partial \mathcal{L}}{\partial \mathbf{y}} \cdot \frac{\partial \mathbf{y}}{\partial \theta_e}\Big|_{\theta_e^t}. \quad (13)$$

Consequently, at each training step, the MoE model simultaneously performs two updates: one optimizes the routing structure $\mathbf{r}(\mathbf{x})$ (via $\theta_r$) to determine which experts are activated and their blending weights, while the other enhances the experts' representation performance (via $\theta_e$).

However, this simultaneous optimization of structure and performance creates inherent interference. At training step $t$, the **observed** expert gradient is

$$\nabla_{\theta_e}^t \mathcal{L} = \frac{\partial \mathcal{L}}{\partial \mathbf{y}} \cdot \sum_{i=1}^{E} \mathbf{r}(\mathbf{x}; \theta_r^t)_i \cdot \nabla_{\theta_{e_i}} \mathbf{f}_i(\mathbf{x}; \theta_{e_i}^t), \qquad (14)$$

where $\mathbf{r}(\mathbf{x}; \theta_r^t) \in \mathbb{R}^E$ is the router output vector under router parameters $\theta_r^t$, and $\mathbf{f}_i(\mathbf{x}; \theta_{e_i}^t)$ denotes the output of the $i$-th expert parameterized by $\theta_{e_i}^t$. In contrast, if training occurred with optimal, stable router parameters $\theta_r^*$, the **ideal** expert gradient would be:

$$\nabla_{\theta_e}^* \mathcal{L} = \frac{\partial \mathcal{L}}{\partial \mathbf{y}} \cdot \sum_{i=1}^{E} \mathbf{r}(\mathbf{x}; \theta_r^*)_i \cdot \nabla_{\theta_{e_i}} \mathbf{f}_i(\mathbf{x}; \theta_{e_i}^t). \qquad (15)$$

Since $\theta_r^t$ continuously evolves during training ($\theta_r^t \neq \theta_r^*$), the observed expert parameter updates constantly deviate from the ideal stable trajectory. This deviation accumulates over time, degrading optimization efficiency. We quantify this accumulated optimization error as the sum of Euclidean distances between the observed and ideal expert gradients:

$$\mathbf{E}_{\text{opt}} = \sum_t \|\nabla_{\theta_e}^t \mathcal{L} - \nabla_{\theta_e}^* \mathcal{L}\|, \qquad (16)$$

which serves as a quantitative measure of interference severity.

To eliminate this error, we propose a decoupled optimization strategy. Rather than simultaneously updating both $\theta_e$ and $\theta_r$ as in (12), we adopt a two-stage approach. First, we obtain near-optimal router parameters $\theta_r^*$ by distilling from a fully trained MoE model (e.g., DeepSeek-V2-Lite, Qwen3-30B-A3B, Qwen3-235B-A22B). Second, we fix $\theta_r^*$ and optimize only the expert parameters using the gradient in (15). This ensures that expert optimization proceeds along the ideal trajectory throughout training.

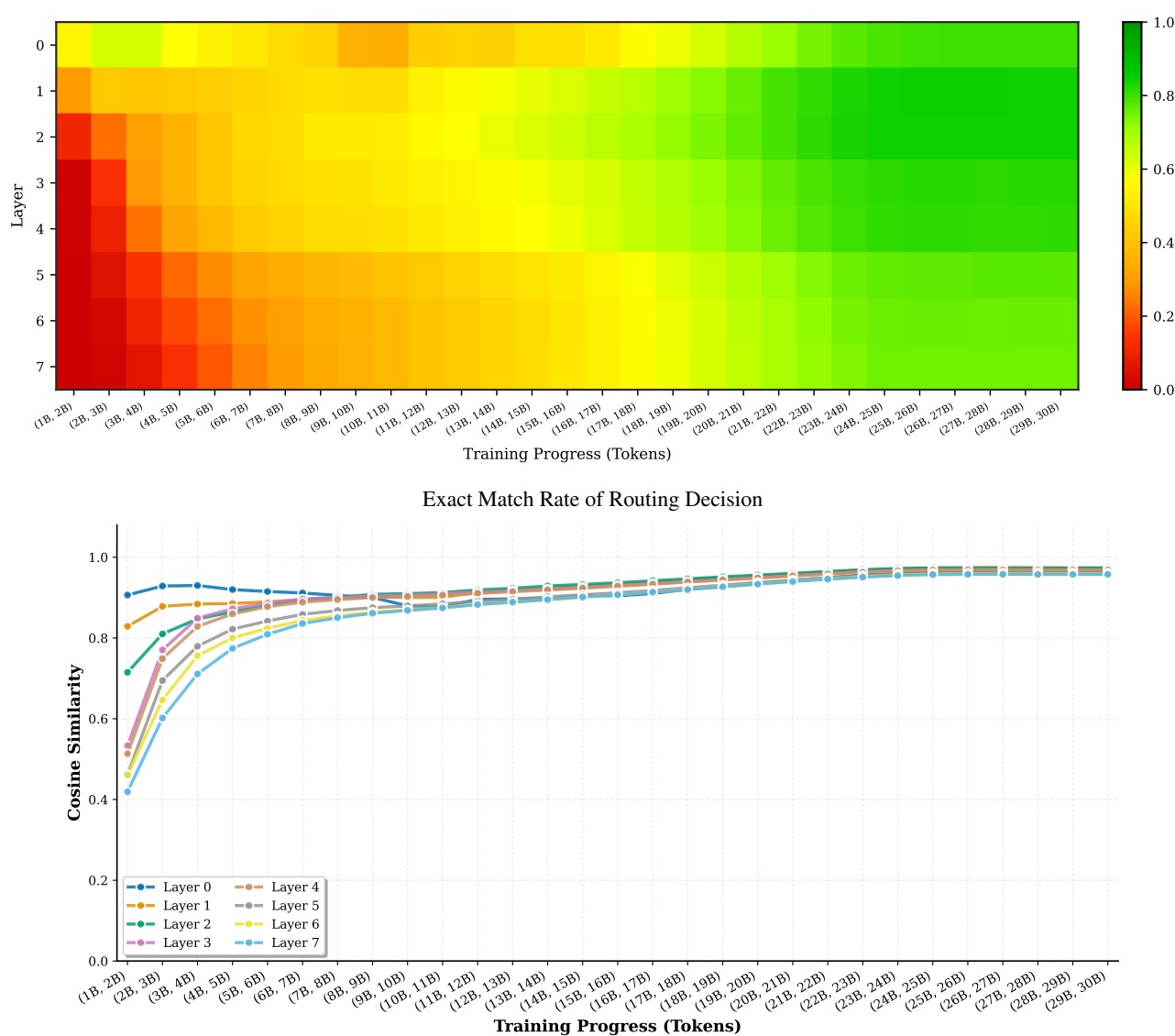

Exact Match Rate of Routing Decision

Cosine Similiarity of Routing Score

*Figure 6.* Routing fluctuations of a 550M model trained on 30B tokens. Given the high token-to-parameter ratio of 60, the model has largely converged. It is observed that the router exhibits intense volatility during the early stages of training; however, as training progresses, the routing patterns gradually converge and maintain stability.

# B. Detailed Desctiption of Routing Fluctuation Heatmap

Figure 6 provides a comprehensive visualization of the routing dynamics throughout the training process. To quantify this evolution, we periodically save model checkpoints and evaluate the routing consistency between adjacent pairs (e.g., at 1B and 2B tokens) by feeding them an identical input batch. Specifically, Figure 6a illustrates the proportion of tokens assigned to the same experts across checkpoints, while Section 6b measures the cosine similarity of routing scores to reflect the magnitude of shift in the router's soft preferences.

As shown in the figures, both expert assignment and routing scores exhibit significant volatility during the early stages of training. However, as training progresses, these two metrics converge toward a steady state, demonstrating increasing routing stability. This observation empirically validates our hypothesis regarding the *Inefficiency of Joint Router-Expert Optimization* discussed in subsection 2.2, confirming that the router eventually reaches an asymptotic equilibrium despite the initial turbulence.

# C. Limitations in Existing Approaches

Although several methods have attempted to decouple the router from the MoE framework, critical challenges remain unresolved. We summarize the limitations of these approaches as follows:

**Limitations in Static Structure Assignment.** Some prior works have attempted to construct stable structures by fixing token-to-expert assignment tables (Roller et al., 2021). However, this approach suffers from several critical drawbacks. First, a fixed token-to-expert mapping contradicts the core design principles of the Transformer. Due to attention mechanisms (Vaswani et al., 2017) and positional encodings (Su et al., 2024), the same token should activate different experts depending on its specific context. A static lookup table eliminates this crucial contextual sensitivity, effectively collapsing the routing space from exponentially many context-dependent assignments to a simple token-identity lookup. This rigidity prevents the model from capturing the nuanced semantic roles a token may play in different sequences. Second, lookup tables are inherently incapable of specifying the continuous blending weights for expert aggregation. While a table can determine which experts are activated, it cannot determine the specific values of $\mathbf{r}(\mathbf{x})$. These weights represent the relative importance of each expert for a given input, which are continuous values that must be computed dynamically by a router. Since these coefficients cannot be reasonably constructed through manual annotation or fixed assignments, a lookup table remains an incomplete solution for the decoupled routing paradigm.

**Limitations in Dynamic Structure Learning.** Compared to static assignment, which can only evaluate structure quality from a load-balancing perspective, learning structure through training is more principled. StableMoE (Dai et al., 2022) exemplifies this approach by constructing a linear network $\mathbf{D}$ that attempts to capture routing structure by distilling the MoE router's output. Specifically, StableMoE divides pretraining into two phases. The first phase employs a triple-loss function:

$$\mathcal{L} = \mathcal{L}_{\text{CE}} + \mathcal{L}_{\text{Bal}} + \mathcal{L}_{\text{Distill}}$$

where $\mathcal{L}_{\text{CE}}$ is the cross-entropy loss, $\mathcal{L}_{\text{Bal}}$ is the load balancing loss, and $\mathcal{L}_{\text{Distill}}$ represents the Kullback-Leibler divergence between $\mathbf{D}$'s predictions and the router's output:

$$\mathcal{L}_{\text{Distill}} = \text{KL}\left(\mathbf{D}(\mathbf{H}_{\text{in}}) \parallel \mathbf{s}(\mathbf{H})\right)$$

During the first phase, the model simultaneously optimizes for task performance, load balance, and distillation. In the second phase, $\mathbf{D}$ is frozen and used to select experts. However, this method faces several critical issues. First, the synchronous optimization of the triple-loss function intensifies the coupling between task performance and structural stability. Forcing the model to simultaneously learn representations, load balancing, and routing distillation creates a volatile optimization landscape that hinders efficient convergence. Second, because model training and distillation occur synchronously, the $\mathbf{D}$ network attempts to learn a continuously evolving target structure, resulting in poor distillation quality. Third, if training collapses or distillation fails, the entire procedure must be restarted from scratch, imposing significant computational costs.

# D. Benefits of Preemptive Structures

Many existing MoE optimization algorithms are fundamentally constrained by their reliance on real-time router outputs during training. These methods must operate after dynamic routing decisions $\mathbf{r}(\mathbf{x})$ are made, limiting optimization to highly restricted conditions. For example, NetMoE (Liu et al., 2025) reduces Expert Parallel (Shazeer et al., 2017) communication costs by computing optimal token placements based on token-to-expert affinity during the dispatch stage. Similarly, FlexMoE (Nie et al., 2023) addresses dynamic load imbalance by adapting hardware resources to tokens' instantaneous routing demands, considering current expert loads and expert-to-GPU mappings. Furthermore, MoE++ (Jin et al., 2024) proposes Zero-Experts, allowing tokens to activate fewer than TopK experts to accelerate inference based on immediate router outputs. Linear-Programming-Based Load Balancer (Cao, 2025) is another method that addresses dynamic load imbalance, yet it still depends on real-time router-derived token indices and per-batch workload statistics to optimize load distribution in

expert-parallel groups, remaining constrained by runtime requirements.

However, the necessity of acquiring router outputs in real-time imposes a "computational bottleneck" on the optimization process itself. This dependency locks the MoE framework into a state of reactive adaptation rather than proactive orchestration. Consequently, establishing a preemptive routing structure—one that is determined before the training loop begins—would unlock a substantially broader optimization space.

## E. Related Works

We discuss additional related work that extends beyond the scope of the main text.

**Large Language Models.** Large Language Models have driven rapid progress across a broad range of tasks, from textual reasoning (Xie et al., 2026; Zhan et al., 2026b) to multimodal understanding and generation (Dong et al., 2025; Zhan et al., 2026a). Modern LLM development typically follows a two-stage paradigm: large-scale pre-training to acquire general knowledge, followed by post-training (Liu et al., 2026) alignment to elicit desired behaviors. Our work addresses the pre-training stage, where MoE architectures have become the dominant approach for scaling model capacity efficiently. While the above advances improve what models learn, GROUTER improves how efficiently MoE models train by providing a structural prior that decouples routing from representation learning.

**Knowledge Distillation.** Knowledge distillation (Hinton et al., 2015) transfers learned knowledge across models of different capacities and has been widely adopted throughout the LLM pipeline, from model compression to policy transfer in RL settings (Yang et al., 2026; Gu et al., 2026a; Qin et al., 2026). GROUTER employs distillation for a distinct purpose: rather than compressing a model's task knowledge, it extracts the routing structure from a converged MoE model into a lightweight standalone network, which then serves as a frozen structural prior for target model training.

## F. The Implementation Details of Baselines

We compare GROUTER against a spectrum of prior works focused on router improvements. All baselines were implemented under the same operating environment and within the same version of the Megatron-LM framework. In addition to **StableMoE** and **HashLayer**, representative methods that address the decoupling problem that we also target, we selected the following representative Router enhancement methods as our baselines: (a) **ReMoE** replaces the Top-K operation with a ReLU gate to mitigate the issue of truncated router gradients. (b) **ToPP** and **MoE++** introduce

mechanisms for dynamically selecting the number of active experts. (c) **Z-Loss** enhances router stability via an additional regularization loss, leading to improved model performance. (d) **Aux-Loss-Free** employs expert bias to remove the auxiliary balancing loss from the optimization objective, thereby avoiding performance degradation caused by loss interference.

All baselines were built upon the Megatron-LM framework (commit hash e7c55de9). For baselines involving specific hyperparameters, we adhere to the following principles: if official code is available, we adopt the original configuration; otherwise, we strictly follow the parameters specified in the respective paper's experimental section.

The specific configurations for each representative baseline are as follows:

- **ReMoE** We use a load balancing loss coefficient of $0.01$ and employ the Pre-Softmax method for determining expert weights, as is standard for this architecture.

- **ToPP** The configuration utilizes a threshold of $0.4$, a load balancing auxiliary loss coefficient of $0.01$, and a dynamic loss coefficient of $0.0001$.

- **MoE++** In addition to the normal experts, we included six constant experts, one zero expert, and one copy expert. We set the auxiliary loss coefficient to $0.01$, the expert capacity factor to $1.1$, and enabled gating residuals.

- **Z-Loss** Following the setup in (Zoph et al., 2022), we set the load balancing loss coefficient to $0.01$ and the Z-loss coefficient to $0.001$.

- **Aux-Loss-Free** We adopt the sigmoid function as the router's normalization function, use a learning rate of $0.001$, and disable the auxiliary loss.

- **HashLayer** We implement the Balanced HashLayer, which showed the best performance in (Roller et al., 2021). Specifically, we first calculate the token frequency distribution in the dataset and then construct the Token-Expert mapping table based on load-balancing criteria. To ensure a fair comparison, since HashLayer inherently lacks a router network for weight assignment, we adapted a router network: during the forward pass, the router network generates logits, the Token-Expert mapping determines the selected expert for each token, and the weights are then calculated via softmax and assigned to the selected experts before continuing the forward propagation.

- **StableMoE** We maintained consistency with (Dai et al., 2022), setting the first pre-training stage to $10\%$ of the total training steps.

*Table 2.* Ablation Study on GROUTER Architectures

| Multi Head Attn | | | | Multi Latent Attn | | | | Residual Linear | |
| --- | --- | --- | --- | --- | --- | --- | --- | --- | --- |
| Decoder | | Encoder | | Decoder | | Encoder | | Linear | |
| Use_Pos | No_Pos | Use_Pos | No_Pos | Use_Pos | No_Pos | Use_Pos | No_Pos | Use_Pos | No_Pos |
| 3.342 | 3.365 | **3.318** | 3.367 | 3.335 | 3.387 | 3.356 | 3.360 | 3.375 | 3.359 |

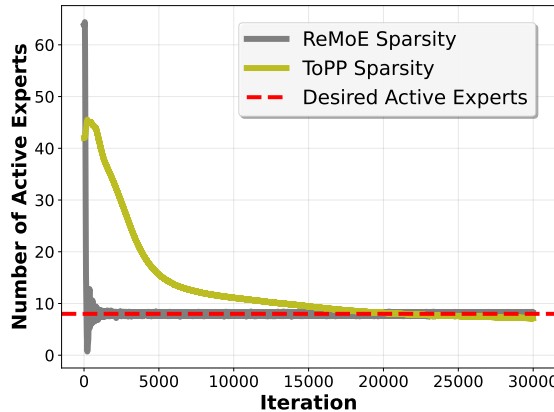

*Figure 7.* Number of Active Experts During Training for Dynamic Activation Baselines. The number of active experts remains consistently higher than the expected value throughout the training process, confirming a fair comparison.

Note that ReMoE and ToPP utilize a setup with a non-fixed number of active experts. We found that under our experimental settings, their number of active experts does not fall below the standard required for a fair comparison. The active expert counts for these methods are visualized in Figure 7.

## G. Ablation

In this section, we conduct ablation studies on several key design choices for the GROUTER. We first focus on the GROUTER's internal structure to determine which architecture is most effective in extracting the model's structural prior. Following prior work on decoupled routers (Dai et al., 2022; Cai et al., 2024), we restrict our scope to three plausible structures: (1) a Multi-Head Attention-based Transformer, (2) a Multi-Latent Attention-based Transformer, and (3) a Linear Layer with Residual Connections. Each of these structures is positioned between the Embedding and the output Linear layers, where it is tasked with capturing the feature structure required for routing in the Source Model.

Beyond the model architecture itself, whether to incorporate positional information is another critical factor. While positional information is not strictly necessary for routing decisions and could potentially introduce noise, its inclu-

sion might enable the GROUTER to better comprehend the Source Model's routing decisions. Consequently, we perform ablations on both the architecture type and the inclusion of positional information.

Specifically, we distill each GROUTER variant for 2.5 Billion tokens and subsequently apply it to the Mini-DS-V2-Lite model for 10 Billion tokens of pretraining. The validation loss results, presented in Table 2, demonstrate that the Multi-Head Attention-based Transformer Encoder exhibits a significant advantage. We attribute this superiority to the Encoder Transformer's inherent attention mechanism, which is better suited to interpret the Source Model's global routing decisions. And its built-in global receptive field allows the GROUTER to leverage more context for token assignment.

We explore how different expert folding methods impact the performance of GROUTER when transferred to models with distinct configurations. We contrast three distinct strategies: Random Folding, where experts are merged arbitrarily; Load-Balance Folding, which merges high-load experts with low-load experts to intrinsically balance the GROUTER's load under the new configuration; and our proposed Affinity-Based Folding, where experts are merged based on their affinity.

We prepared three Grouter variants, one for each folding method. Subsequently, these variants were applied to the Mini-GPT-OSS model and pre-trained on 5 Billion tokens. The results, measured by the final validation set loss, are presented in Figure 9. The comparison clearly shows that the affinity-based expert folding yields the best performance. This is because high expert affinity indicates a greater shared functionality; therefore, merging these experts minimizes the loss of crucial structural information, leading to better model fidelity and overall performance.

## H. Maximal Violation

MaxVio_{Global} serves as a robust and widely used indicator of the global load balance and is formally defined as:

$$\text{MaxVio}_{\text{Global}} = \frac{\max_i L_{e_i} - \overline{L}}{\overline{L}} \qquad (17)$$

where $L_{e_i}$ is the total number of tokens afforded to expert $e_i$ across the entire training process, and $\overline{L}$ represents the uniform average load distributed among all experts.

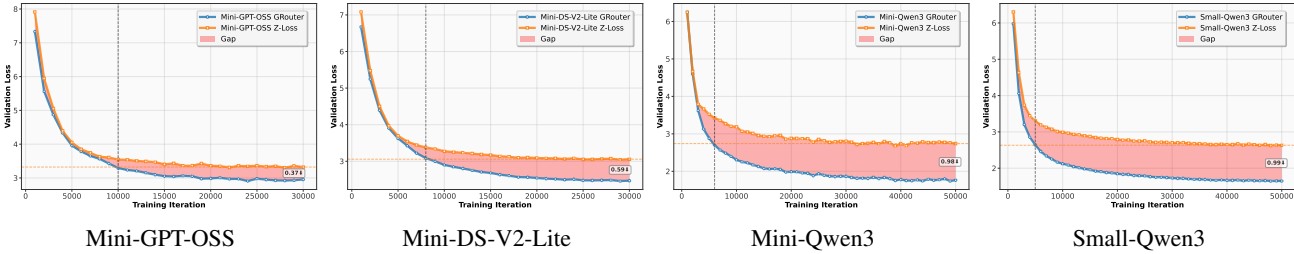

Mini-GPT-OSS          Mini-DS-V2-Lite          Mini-Qwen3          Small-Qwen3

*Figure 8.* Training curves of Grouter and Z-loss for models with varying sizes, architectures, and expert counts. The results demonstrate that a single Grouter can be migrated to different configurations via expert folding and tuning, while retaining superior convergence speed and precision.

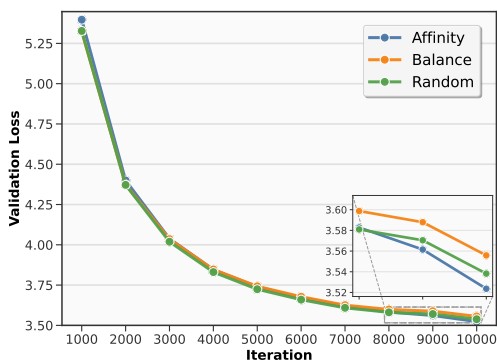

*Figure 9.* Ablation Study on Expert Folding Strategies.

## I. Downstream Experiments

We selected the standard evaluation datasets: BoolQ (Clark et al., 2019), RTE (Wang et al., 2018), HellaSwag (Zellers et al., 2019), PIQA (Bisk et al., 2020), RACE (Lai et al., 2017), and MNLI (Williams et al., 2017).

As the Mini-Qwen3 model we trained is a Base version without instruction-following fine-tuning, we adopted a Perplexity (PPL) ranking strategy for downstream task evaluation. Specifically, for each task, we construct prompts by concatenating the context and different options. These prompts are fed through the model's forward pass to obtain the PPL for each option. The option that yields the lowest PPL is selected as the model's prediction and compared against the ground truth.

The specific prompt templates used for each task are detailed below:

**Prompt Templates** The specific prompt templates used for the PPL ranking strategy are detailed below. We use the `Context/Premise` and `Question/Hypothesis` fields from each dataset to construct the final prompt. For tasks marked with an asterisk (*), the final answers ("Yes" / "No") are appended for PPL calculation and ranking.

**BoolQ** `Context is {passage}\nQuestion is {question}\nDoes the context answer`

the question with 'Yes'?  Answer
with 'Yes' or 'No':

**RTE** `Premise is {premise}\nHypothesis is {hypothesis}\nDoes the premise entail the hypothesis?  Answer with 'Yes' or 'No':`

**HellaSwag*** `Context is: {context}\nEnding is: {ending}\nIs this ending a reasonable continuation of the context?  Answer with 'Yes' or 'No':`

**PIQA*** `Goal is: {goal}\nSolution is: {sol}\nIs this solution effective for the goal?  Answer with 'Yes' or 'No':`

**RACE*** `Article is {article}\nQuestion is {question}\nIs the option '{option}' the correct answer to the question?  Answer with 'Yes' or 'No':`

**MNLI** `Premise: {premise}\nHypothesis: {hypothesis}\nWhat is the relationship between the premise and hypothesis?  Choose from 'entailment', 'neutral', 'contradiction':`

Note on Binary Tasks: For HellaSwag, PIQA, and RACE, the full prompt includes appending the target answers, such as " Yes" or " No," to the base template for PPL calculation and ranking.

## J. Efficiency Experiment Detail

In this section, we present the detailed experimental settings for the efficiency evaluations.

**Pre-Dispatch.** We initially employ the data processing pipeline of Megatron-LM to tokenize and pack the raw

data, thereby transforming the pre-training dataset into fixed-length training sequences. Concurrently, we process these generated sequences using GROUTER. To maximize storage efficiency, we pre-compute and store the expert assignment indices in `uint8` format and the routing scores in `bfloat16` format, which significantly reduces the storage footprint.

**Expert Grouping.** For each sequence, we compute routing affinity vectors to serve as the basis for clustering. We employ a hybrid clustering strategy: initially partitioning all sequences into 100 clusters using K-means++ (Arthur & Vassilvitskii, 2006), followed by Agglomerative Hierarchical Clustering (Hastie, 2009) to further merge these into targeted final clusters. Prior to clustering, we filter out sequences with uniformly distributed routing affinity vectors. Such sequences exhibit no distinct preference for specific expert groups; removing them reduces computational overhead without compromising the grouping structure. In our experiments, this is implemented by discarding sequences where the routing affinity vector's entropy exceeds 6.85. Upon completing the clustering, we formulate a bipartite matching problem where the two sets of nodes represent the sequence clusters and the experts, respectively. The edge weights are defined by the average routing affinity vector of each cluster. We solve this problem using the Hungarian algorithm to establish $N$ fixed expert groups, where $N$ depends on the Expert Parallel size and the optimization granularity. We consider two granularity levels: GPU-level and Node-level. Specifically, for a EP group, $N$ equals the total count of individual GPUs for GPU-level optimization, or the total count of server nodes for Node-level optimization. When operating at the node level, experts mapped to a node are subsequently distributed randomly across its intra-node GPUs.

**Sample Placement Optimization.** We perform sequence assignment based on the specialized communication pattern of DeepEP (Liu et al., 2024). Specifically, for each sequence, we calculate the potential communication volume associated with its placement on different candidate devices. We implemented the Algorithm 1 using the NumPy library to facilitate rapid sequence assignment.

## K. Training curves under different configurations

We present the detailed experimental curves for the models under the four different setups shown in Appendix L in Figure 8. The results demonstrate that the advantages of Grouter persist across different configurations.

As illustrated by the results, the performance advantage of GROUTER expands with increasing model scale. Table 3 details the breakdown of total versus expert-specific param-

---

**Algorithm 1** Optimize Sample-to-Node Assignment

---

**Require:** $\mathcal{S}$: Set of samples, $\mathcal{E} : \mathcal{N} \mapsto 2^{\mathbb{N}}$: Expert placement map, $k$: Number of experts per token, $N$: Total number of nodes

1: Initialize $\mathcal{A} = \{n \mapsto \emptyset \mid n \in \{0, 1, \ldots, N-1\}\}$
2: **for all** $s \in \mathcal{S}$ **do**
3:      best_node $\leftarrow 0$, max$_{\text{comm}} \leftarrow -\infty$
4:      $\mathbf{D}_s \leftarrow \text{reshape}(\text{dispatch}(s), (-1, k))$
5:      **for all** $n \in \mathcal{N}$ **do**
6:          $\mathcal{T}_s(n) \leftarrow \{t \in \mathcal{T}_s \mid \mathbf{D}_s[t,:] \cap \mathcal{E}(n) \neq \emptyset\}$
7:          comm$(n, s) \leftarrow |\mathcal{T}_s(n)|$
8:          **if** comm$(n, s) >$ max$_{\text{comm}}$ **then**
9:              max$_{\text{comm}} \leftarrow$ comm$(n, s)$
10:             best_node $\leftarrow n$
11:          **end if**
12:      **end for**
13:      $\mathcal{A}(\text{best\_node}) \leftarrow \mathcal{A}(\text{best\_node}) \cup \{s\}$
14: **end for**
15: **return** $\mathcal{A}$

---

eters across the three architectures. A key observation is that the ratio of expert parameters to total parameters scales positively with model size. Given that the primary mechanism of GROUTER is to accelerate convergence by fostering expert specialization, an increased proportion of expert parameters amplifies its efficacy, resulting in more substantial performance gains. We reserve a more granular analysis of this scaling behavior for future research.

*Table 3.* Parameter statistics for the evaluated models

| Model | Mini-GPT-OSS | Mini-DS-V2-Lite | Mini-Qwen3 |
|---|---|---|---|
| Parameter | 315M | 958B | 2.81B |
| Expert Parameter | 199M | 785M | 2.41B |
| Expert Ratio | 63.19% | 81.99% | 85.96% |

## L. Configuration

The detailed configurations for all MoE models employed in our experiments are presented in Table 4, covering key specifications (e.g., parameter count, hidden dimension, expert setup) for models with diverse scales and architectural designs to support reproducibility and cross-reference in this paper. **Note that Small-Qwen3 is omitted from this table, as it is derived by simply doubling the expert count of Mini-Qwen3 to 256 to form a 7B model.**

## M. The distillation Loss of GROUTER

The convergence curve, presented in Figure 10, confirms that the GROUTER effectively learns the structural prior from the Source Model, reaching a stable, low loss value.

*Table 4.* Model Configurations

| Model | Mini-GPT-OSS | Tiny-Qwen3 | Mini-DS-V2-Lite | Mini-Qwen3 |
|---|---|---|---|---|
| Parameter | 350M | 550M | 1B | 3B |
| Hidden_Size | 720 | 384 | 512 | 1024 |
| Num_Layer | 8 | 8 | 12 | 16 |
| Num_Experts | 32 | 128 | 64 | 128 |
| Top_k | 4 | 8 | 6 | 8 |
| FFN_Hidden_size | 720 | 688 | 2736 | 3072 |
| Num_Attention_Heads | 32 | 8 | 8 | 16 |
| MoE_FFN_Hidden_size | 720 | 344 | 704 | 384 |
| Max_Position_Embeddings | 131072 | 32768 | 4096 | 40960 |

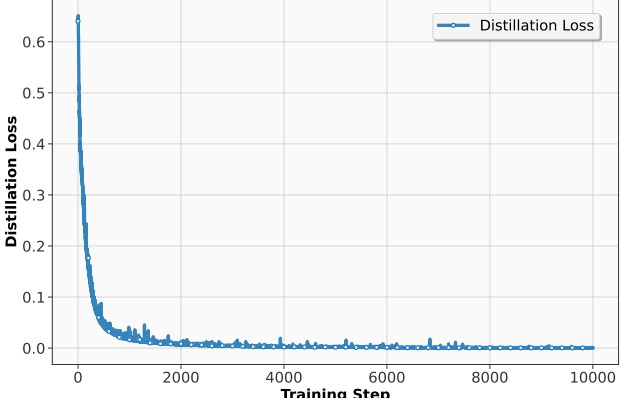

*Figure 10.* Learning Curve of the Distillation Process

# N. Discussion on the Efficient Convergence Achieved by Grouter

We illustrate the coefficient of variation of the gradient norm under different sliding window sizes in Figure 11a, Figure 11b, effectively demonstrating stability across both short and long timescales. Regardless of the temporal scale, GROUTER exhibits exceptional stability; the gradient norm remains consistently smooth without abrupt fluctuations or spikes. This not only confirms the robust training stability of GROUTER but also indicates its effectiveness in ensuring that experts consistently receive familiar tokens, thereby facilitating continuous specialization throughout the training process.

Figure 11c depicts the training gradient norms, showing that GROUTER consistently yields larger gradient magnitudes. We analyze this by considering the aggregated gradient $\mathbf{G} = \sum_{i \in \mathcal{B}} \mathbf{g}_i$ for an expert over a batch of assigned tokens $\mathcal{B}$. The squared norm of this gradient is given by:

$$\|\mathbf{G}\|^2 = \sum_{i \in \mathcal{B}} |\mathbf{g}_i|^2 + \sum_{i \neq j} \mathbf{g}_i^\top \mathbf{g}_j \quad (18)$$

where $\mathbf{g}_i$ represent the gradient of token i.

In GROUTER, the pre-learned structure routes semantically similar embeddings to the same expert, resulting in positively aligned gradients where $\mathbf{g}_i^\top \mathbf{g}_j > 0$. This constructive interference significantly amplifies the total gradient magnitude $\|\mathbf{G}\|$. Conversely, traditional routers often assign

heterogeneous embeddings to the same expert due to the lack of priors. This leads to conflicting optimization directions where $\mathbf{g}_i^\top \mathbf{g}_j < 0$ for many pairs, causing gradient cancellation that diminishes the effective update magnitude.

As training progresses, the update of expert parameters may stagnate when the following condition is met:

$$\sum_{i \neq j} \mathbf{g}_i^\top \mathbf{g}_j \approx -\sum_{i \in \mathcal{B}} |\mathbf{g}_i|^2 \quad (19)$$

When this occurs, the aggregated gradient vanishes due to cancellation. Consequently, the router tends to freeze its current state, creating an illusion of convergence. In reality, however, the model retains significant capacity for optimization; further progress is merely impeded by the conflicting representational requirements of diverse embeddings. By optimizing the routing structure to assign semantically similar embeddings to specific experts, GROUTER effectively mitigates these conflicts and delays the onset of the condition defined in (19). This mechanism significantly extends the model's optimization trajectory. It also explains why GROUTER maintains a substantial gradient norm even when baseline methods approach zero—a finding that aligns with the superior convergence potential observed in our experiments.

# O. Cross-Scale Transferability of GROUTER

A key question for the practical applicability of GROUTER is whether its structural prior transfers across model scales—specifically, whether a GROUTER distilled from a smaller source model can effectively guide the training of a larger target model ($P_T > P_S$), or vice versa ($P_S > P_T$). We investigate the former scenario, as it is particularly relevant in practice: practitioners may only have access to a small, trained MoE model yet wish to leverage its routing structure to accelerate the training of a larger target model.

**Experimental Setup.** Due to the limited availability of small open-source MoE models, we first pre-train a 550M-parameter Tiny-Qwen3 on 30B tokens. We then distill GROUTER from its first MoE layer and apply this GROUTER to guide the training of Mini-DS-V2-Lite. This setup constitutes a challenging cross-architecture, cross-scale transfer: the source and target models differ in both parameter count and expert configuration.

**Results.** Figure 12 presents the validation loss curves over 20B tokens pre-training. Despite being distilled from a substantially smaller model, GROUTER achieves performance comparable to the baseline throughout training.

These results demonstrate that GROUTER's structural prior generalizes across model scales. Combined with the throughput gains from preemptive routing, this cross-scale

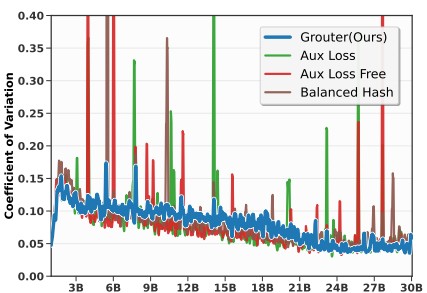 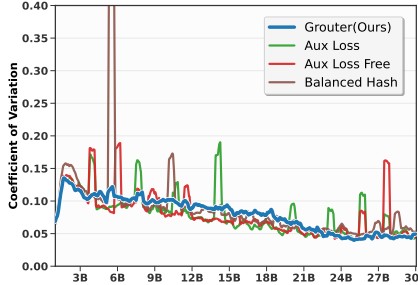 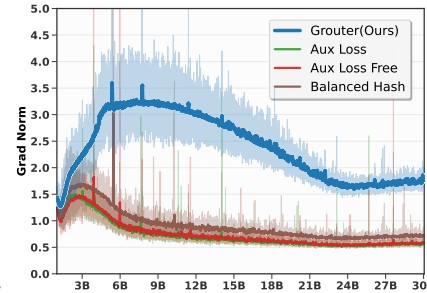

*Figure 11.* Analysis of training stability and gradient magnitudes. (a) The CV of the gradient norm, computed using a sliding window of size 50. This illustrates Grouter's superior stability at short timescales, whereas baseline methods exhibit significant spikes. (b) The gradient norm CV with a larger sliding window size of 500, demonstrating Grouter's sustained stability over longer timescales. (c) The trajectory of the gradient norm throughout training. An Moving Average curve (window size 50) is highlighted for visual clarity. The results indicate that Grouter consistently maintains a higher gradient norm compared to baselines.

transferability translates directly into significant end-to-end training acceleration—even when the GROUTER is distilled from a model substantially smaller than the target.

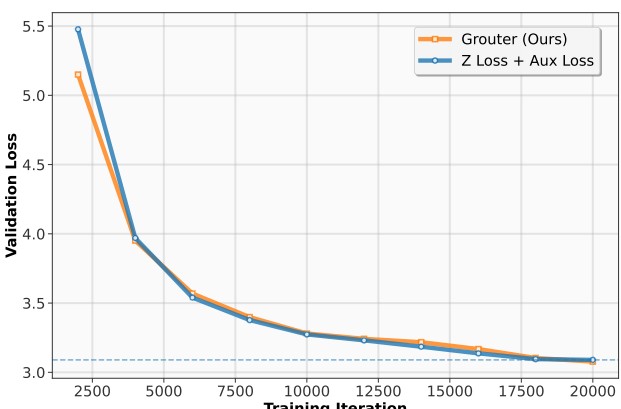

*Figure 12.* Validation loss comparison for cross-scale transfer. GROUTER matches or even outperforms the baseline throughout, demonstrating effective cross-scale structural transfer.

## P. Overhead Analysis of Preemptive Routing

A natural concern regarding preemptive routing is its additional overhead in storage and computation. We provide a detailed cost breakdown below, demonstrating that the total overhead of the GROUTER pipeline is modest relative to the substantial training savings it enables.

**Storage Requirements.** During preemptive routing, the routing decisions for each token are cached to disk. We store each activated expert index as a `uint8` value (1 byte) and its corresponding routing score in `bfloat16` (2 bytes), yielding a per-token storage cost of $k \times 3$ bytes, where $k$ denotes the number of activated experts. For Qwen3 with $k=8$, this amounts to 24 bytes per token. For a pretraining

corpus on the order of trillions of tokens, the total storage requirement is approximately tens of terabytes. This is well within the petabyte-scale capacity of modern cluster storage systems. Moreover, this storage need not be allocated simultaneously: routing decisions can be computed and consumed incrementally, with completed batches deleted before processing subsequent data.

**Computational Requirements.** We further analyze the computational cost of Mini-Qwen3. We estimate FLOPs following the standard approximation from Brown et al. (2020): for a Transformer with $l$ layers and hidden dimension $h$, the parameter count is approximated as $P \approx 12lh^2$, and the computational costs for $N$ tokens are $C_{\text{fwd}} = 2NP$ (forward) and $C_{\text{bwd}} = 4NP$ (backward). For MoE models, we apply a sparsity correction factor $\alpha = E_a/E$, where $E_a$ and $E$ denote the number of activated and total experts, respectively. The relevant model configurations are summarized as follows:

- **GROUTER:** $l_{\mathbf{G}} = 3$, $h_{\mathbf{G}} = 512$.
- **Source model (Qwen3-MoE):** $h_{\mathbf{S}} = 2048$, $\alpha_{\mathbf{S}} = 1/16$.
- **Target model (Mini-Qwen3):** $l_{\mathbf{T}} = 16$, $h_{\mathbf{T}} = 1024$, $\alpha_{\mathbf{T}} = 1/16$.

We now detail the cost of each pipeline stage.

**Distillation.** During distillation, each training sequence requires one forward pass through both GROUTER and the source model, plus one backward pass through GROUTER. Due to the sequential nature of Transformers, only the first $n=1$ layer of the source model needs to be executed. Using $N_d = 2.4$B distillation tokens:

$$C_{\text{distill}} = N_d \big( 6 \cdot 12 l_{\mathbf{G}} h_{\mathbf{G}}^2 + 2 \cdot 12 \cdot h_{\mathbf{S}}^2 \cdot \alpha_{\mathbf{S}} \big)$$
$$\approx 150{,}995 \text{ TFLOPs.} \tag{20}$$

**Expert Folding** (Optional). Computing the expert co-activation affinity matrix requires a single GROUTER forward pass over $N_f = 52\text{M}$ tokens:

$$C_{\text{fold}} = N_f \cdot 2 \cdot 12l_{\mathbf{G}}h_{\mathbf{G}}^2 \approx 981 \text{ TFLOPs.} \tag{21}$$

**Expert Tuning.** Fine-tuning the final projection layer for load balance requires forward and backward passes through GROUTER on $N_t = 52\text{M}$ tokens:

$$C_{\text{tune}} = N_t \cdot 6 \cdot 12l_{\mathbf{G}}h_{\mathbf{G}}^2 \approx 2{,}944 \text{ TFLOPs.} \tag{22}$$

**Preemptive Routing.** Pre-computing routing decisions for $N_r = 6\text{B}$ tokens requires a single GROUTER forward pass:

$$C_{\text{route}} = N_r \cdot 2 \cdot 12l_{\mathbf{G}}h_{\mathbf{G}}^2 \approx 113{,}246 \text{ TFLOPs.} \tag{23}$$

**Pretraining with GROUTER.** With the precomputed structural prior, Mini-Qwen3 achieves equivalent loss with only $N_g = 6\text{B}$ tokens:

$$\begin{aligned} C_{\text{train}}^{\text{GROUTER}} &= N_g \cdot 6 \cdot 12l_{\mathbf{T}}h_{\mathbf{T}}^2 \cdot \alpha_{\mathbf{T}} \\ &\approx 452{,}983 \text{ TFLOPs.} \end{aligned} \tag{24}$$

**Baseline (Pretraining Directly).** Pretraining Mini-Qwen3 from scratch to the same loss requires $N_0 = 50\text{B}$ tokens:

$$C_{\text{train}}^{\text{base}} = N_0 \cdot 6 \cdot 12l_{\mathbf{T}}h_{\mathbf{T}}^2 \cdot \alpha_{\mathbf{T}} \approx 3{,}774{,}873 \text{ TFLOPs.} \tag{25}$$

The complete breakdown is presented in Table 5.

*Table 5.* Computational cost breakdown of the GROUTER pipeline compared with standard pretraining. Percentages are relative to the baseline pretraining cost.

| Stage | Cost | Relative Cost |
|---|---:|---:|
| Distillation | 150,995 | 4.00% |
| Expert Folding | 981 | 0.03% |
| Expert Tuning | 2,944 | 0.08% |
| Preemptive Routing | 113,246 | 3.00% |
| Pretraining w/ GROUTER | 452,983 | 12.00% |
| **Total w/ GROUTER** | **721,149** | **19.11%** |
| Pretraining Directly | 3,774,873 | 100.00% |

The total GROUTER pipeline cost amounts to approximately 721,149 TFLOPs, constituting only **19.1%** of the baseline pretraining cost. As shown in Table 5, the additional overhead introduced by the GROUTER pipeline—encompassing distillation, expert folding, expert tuning, and preemptive routing—accounts for only approximately $7.1\%$ of the baseline pretraining cost. The dominant cost reduction stems from the improved sample efficiency: the target model achieves equivalent loss with $8.3\times$ fewer training tokens.

We further note two practical considerations that reduce the effective overhead. First, GROUTER can be obtained directly from open-source releases, entirely eliminating the distillation cost for practitioners who do not train their own. Second, preemptive routing can be scheduled during GPU idle periods prior to the main training run, effectively overlapping with other cluster operations at zero marginal wall-clock cost. Additionally, the expert grouping and sample placement optimization stages are performed entirely on CPUs using standard scientific computing libraries. Given that CPU resources in modern training clusters are typically underutilized during GPU-intensive workloads, these stages introduce negligible wall-clock overhead when executed concurrently with GPU training preparation.

