# OpenReview forum: "Grouter: Decoupling Routing from Representation for Accelerated MoE Training"
_ICML.cc/2026/Conference — ICML 2026 regular_

### Official Review · Reviewer_ubHS · 2026-02-28

**Soundness:** 3
**Presentation:** 2
**Significance:** 3
**Originality:** 3
**Overall Recommendation:** 3
**Confidence:** 3

**Summary:**

- The paper introduces GROUTER, a method designed to accelerate Mixture of Experts (MoE) training by decoupling routing structure learning from expert representation learning.
- It distills a fixed routing structure from a fully converged, large-scale source MoE model into a lightweight, standalone router model.
- To adjust different characteristics between the source MoE model and a target MoE model, the authors propose Expert Folding to map source experts to target experts using an affinity matrix, and Expert Tuning to adjust for target data distribution shifts.
- Fixed structural prior enables offline preemptive routing, shifting communication optimization for Expert Parallelism (EP) to a preprocessing step.
- Empirical results demonstrate up to a 4.28x improvement in data efficiency during pre-training, and up to a 33.5% throughput acceleration compared to dynamic routing baselines.

**Compliance With Llm Reviewing Policy:**

Affirmed.

**Final Justification:**

Grouter accelerates Mixture of Experts (MoE) model training drastically, saving a lot of cost in training further models. After rebuttal, the reviewers introduce expert expanding strategy to resolve the limitation of existing Grouter that was only able to train smaller model. Grouter is generally applicable using various existing MoE models. They fully address concerns about overhead and feasibility of training larger target model, which are not initially introduced but required to have an additional Grouter component in training.

In terms of the design, its practicality, and its usefulness, Grouter should be accepted. However, the remaining concern in writing and representation that cannot be addressed in rebuttal makes me stay in negative score, even though I have raised my score from the initial rate.
More specifically,
1. The authors acknowledge that the design section needs refinement.
2. The authors introduce a new design -- expert expansion -- in rebuttal and is not included in the initial main body.

Applying these two changes into the body should change the paper structure drastically, which does not ensure quality of the paper.

**Key Questions For Authors:**

- **Parameter scaling limits**: Can GROUTER successfully guide the training of a target model that is significantly larger than the source model used for distillation? If the target model has vastly more capacity, does the structural prior of a smaller teacher artifically cap the targe'ts final accuracy?
- **Expert upscaling**: the expert folding groups source experts with an average size $G_{\text{avg}}=\lfloor E_S / E_T \rfloor$. How does the framework handle target architectures with a larger number of experts than the source model ($E_T > E_S$)?
- **Layer-specific distillation and stability: the text claims the first MoE layer is the most stable, while Figure 6 shows a counter example. Given this discrepancy, and the likelihood that routing dynamics are architecture-specific, how robust is the hueristics of always distlling from layer 0?
- **End-to-end computational overhead**: Appendix E notes that the GROUTER has 60M parameters and is distilled using 2.6B tokens. While 2.6B tokens is a small fraction of a typical pre-training token budget, running a 30B source MoE over 2.6B tokens for distillation, fine-tuning for expert folding and tuning, and communication optimization that requires expert preference vectors carry some computational cost. Could the authors provide a quantitative, end-to-end wall-clock or FLOPS breakdown of this distillation preprocessing overhead versus the net savings achieved during the target MoE training phase?

**Limitations:**

- The paper lacks any evaluation or discussion of scenarios where the target model is larger than the teacher model. It remains unclear if a structural prior from a smaller model contains sufficient semantic granularity to effectively optimize a larger parameter space.
- Offline routing requires caching indices and scores for the entire dataset. The paper lacks a dicussion on the computational cost and storage space requirements for trillion-token scales.
- The paper assumes layer 0 is universally the best starting point, but lacks evidence that this stability pattern holds across different MoE architectures (e.g. DeepSeek vs Qwen).

**Strengths And Weaknesses:**

$+$ The analysis of routing representation interference is logically grounded, and the empirial demonstration of routing instability early in training effectively motivates the decoupling approach.

$+$ While knowledge distillation typically transfers representational capabilities, using it to extract structural priors represents a creative paradigm shift in MoE optimization.

$+$ Improving convergence speed by 4.28x are substantial practical contributions for teams training MoE models.

$-$ The system overview provided in Figrue 2 lacks sufficient textual explanation to establish a clear timeline of operations. Figure 2(a) indicates the GROUTER is frozen during target MoE training, while Figrue 2(b) illustrates Expert Folding and Expert Tuning as a fine-tuning step. It is not clear how such intermediate preparation phase occurs prior to the main target MoE training.

$-$ There is an empirical gap regarding model scaling. The experiments only validate GROUTER on small target models (up to 3B parameters) that are an order of magnitude smaller than the 30B teacher model used for distillation. It remains unproven whether the framework can scale upward. Expert folding is also constrained to downsizing (works only when $E_T <= E_S$), leaving the framework unequipped to handle expert upscaling.

$-$ There is a contradiction between the text's justification for distilling from the first source MoE layer (layer 0) and the empirical evidence in Figure 6. While the text claims the first layer is the most stable, Figure 6 indicates that intermediate layers (indices 1 and 2) achieve higher stability after extensive training. The framework also assumes this layer 0 stability hueristic generalizes across all architectures, which lacks cross-model ablation.

$-$ Section 3.4 introduces hardware-level communication optimization for Expert Parallelism (EP) explicitely referring to this as "another key innovation". However, this hardware optimization is entirely absent from the core contributions listed in the Introduction. Because the paper's primary focus is decoupling routing from representation, Section 3.4 feels disconnected from the core design narrative and distracts from the main text. Acknowledging decoupling design allows this optimization, this content may be better suited for an Appendix to maintain focus.

---

> ### Author Rebuttal · Authors · 2026-03-31
>
> We sincerely thank Reviewer for the insightful and rigorous feedback.
> 1. On Model Scaling and Upward Compatibility (Addressing Question 1 & Weakness 2)
>
> Due to space limitations, please refer to the first point in our response to Reviewer mByH.
>
> 2. On Expert Upscaling ($E_T > E_S$) and New Architecture Support (Addressing Question 2 & Weakness 1 & Limitation 1)
>
> Due to space limitations, please refer to the second point in our response to Reviewer Jir5.
>
> 3. On Layer-0 Stability and Cross-Architecture Robustness (Addressing Weakness 3 & Question 3 & Limitation 3)
>
> Our Response: Clarification on Fig 6:
>
> Our choice of which layer to distill is not based solely on stability metrics, but rather on a dual consideration of signal integrity and computational efficiency:
>
> - Error Propagation in Routing: Routing distributions in deeper layers are inherently more volatile as they are compounded by the fluctuations of all preceding layers. This suggests that earlier layers capture a "cleaner" and less-disturbed structural prior, which is more suitable for guiding a target model from scratch.
> - Hardware and Computational Efficiency: Due to the sequential nature of Transformers, obtaining the routing logits for the $n^{th}$ layer only requires a forward pass through the first $n$ layers. By distilling the earliest possible MoE layer (e.g., Layer 1), we drastically reduce memory and compute requirements, as the remainder of the massive source model (e.g., layers $n+1$ to 32) does not need to be loaded or executed. Our implementation fully leverages this to ensure a lightweight distillation process.
> Figure 6 effectively supports our argument that during training, fluctuating routing in routers of earlier layers continuously disturbs subsequent layers, leading the latter routers to learn more volatile routing patterns. If the router in the first layer keeps fluctuating, the input distribution received by the second router will change constantly, preventing it from learning a reasonable assignment strategy.
>
> [Cross-Architecture Validation (DeepSeek-V2)]: To further demonstrate the robustness of this strategy across diverse architectures, we distilled a Grouter from DeepSeek-V2-Lite  to assist in training a Mini-DSV2L target model. Since Layer 0 of DeepSeek-V2 is a dense layer, we selected Layer 1 (the first MoE layer) for distillation. The validation loss comparison during the first 10B tokens of pre-training is presented below
>
>
> Validation loss comparison on Mini-dsv2lite with and without Grouter distilled from deepseek-v2-lite.
> | Method / Token | 1B | 2B | 3B | 4B | 5B | 6B | 7B | 8B | 9B | 10B |
> | --- | --- | --- | --- | --- | --- | --- | --- | --- | --- | --- |
> | ZL | 5.086 | 3.906 | 3.590 | 3.471 | 3.378 | 3.332 | 3.306 | 3.289 | 3.259 | 3.265 |
> | Grouter-dsv2lite | 4.734 | 3.768 | 3.479 | 3.256 | 3.129 | 3.050 | 2.994 | 2.982 | 2.963 | 2.946 |
>
>
>
> 4. On End-to-End Overhead Analysis for Grouter Method (Addressing Question 4)
>
> Due to space limitations, please refer to the third point in our response to Reviewer Jir5.
>
>
> 5. Storge and Computation Overhead Analysis for Preemptive Routing (Addressing Limitation 2)
>
> We utilize uint8 to store the expert IDs for each token and bf16 for their corresponding routing scores. This results in a storage requirement of approximately $E_{act} \times 3$ bytes per token (where $E_{act}$ is the number of active experts). For the Qwen3 configuration ($E_{act}=8$), this equates to 24 bytes per token, or 24TB for a 1T-token dataset.While 24TB is a non-trivial volume, it remains well within the capacity of modern industrial clusters, which typically measure storage in Petabytes (PB). Furthermore, this storage requirement does not need to be fulfilled concurrently. In practice, pre-dispatch metadata can be processed and deleted in streaming batches, significantly reducing the active storage footprint throughout the training lifecycle.
>
>
> 6. On Presentation and System Scope (Addressing Weakness 1 & 4)
>
> We acknowledge that the presentation of the timeline needs refinement.
>
> Timeline Clarification: Grouter Distillation --> Expert Folding(if $E_T \neq E_S$) --> Expert Tuning --> Training.
>
> Integration of Section 3.4: We respectfully but strongly disagree with the reviewer’s claim that out preemtive routing (section 3.4) optimization is "entirely absent" from our Introduction.
>
> In fact, we explicitly highlighted this in our third core contribution (Page 2 Line 88), stating that by 'Leveraging the fixed priors provided by Grouter, we shift data optimization from runtime to a pre-processing stage.'

---

> > ### Author Rebuttal · Reviewer_ubHS · 2026-04-02
> >
> > Thank you for thorough response. Regarding absency of Section 3.4 in introduction, I acknowledge that the third contribution in Introduction does refer to Section 3.4. Thank you for the clarification. As most of my concerns are resolved, I would raise my score.
> >
> > However, my weakness 2 and question 1 does not appear to be fully resolved. Could you provide any result with models larger than 3B parameters? I can only see the result about training 1B model with grouter distilled from 550M model from the response to Reviewer mByH.

---

> > > ### Author Response · Authors · 2026-04-04
> > >
> > > Thank you for your constructive feedback and for recognizing the clarifications in our response. We are encouraged that most of your concerns have been resolved.
> > >
> > > # On the Weakness 2 and Question 1
> > >
> > > ## For Question 1
> > > In Upward Scaling (the first part of our first response to Reviewer mByH), we conducted experiments to address the reviewer's concern:
> > >
> > > > Can GROUTER successfully guide the training of a target model that is significantly larger than the source model used for distillation? If the target model has vastly more capacity, does the structural prior of a smaller teacher artifically cap the targe'ts final accuracy?
> > >
> > > Specifically, we utilized the GROUTER derived from a 550M model to assist in training a 1B target model(**target MoE is twice as large as the source MoE**). Our results demonstrate that even with a training budget comparable to the source MoE (20B tokens vs. 30B tokens), we consistently achieve comparable loss values. This confirms that using GROUTER to guide a larger target MoE is highly feasible and does not impose an artificial ceiling on the target MoE's performance potential.
> > >
> > >
> > > ## For Weakness 2
> > > Due to space constraints, we noted in the upward scaling section (second part of first response to Reviewer mByH)
> > >
> > > >relevant experimental results are detailed in our second response to Reviewer Jir5.
> > >
> > > in there (second response to Reviewer Jir5) we conducted pre-training experiments using a **7B** model, which effectively validates the acceleration capabilities of Grouter for larger model.
> > >
> > > ### The Summary of model scaling experiment
> > >
> > > To provide a clearer demonstration of our findings, we have summarized the previous **7B** model experiments and incorporated new results from a **16B** model experiment. The Grouter we used is exactly the same as the one employed in the paper.
> > >
> > > - **7B MoE Performance**: We validated Grouter's performance on a 7B model using the expert expansion method. The results are as follows:
> > >
> > > | Method / Token | 1B | 2B | 3B | 4B | 5B | 6B | 7B | 8B | 9B | 10B |
> > > | --- | --- | --- | --- | --- | --- | --- | --- | --- | --- | --- |
> > > | ZL | 6.332 | 4.626 | 3.729 | 3.459 | 3.306 | 3.199 | 3.134 | 3.076 | 3.032 | 2.995 |
> > > | Grouter | **6.055** | **4.260** | **3.364** | **2.864** | **2.595** | **2.398** | **2.273** | **2.189** | **2.101** | **2.065** |
> > >
> > > - **16B MoE Performance**: To further isolate variables and showcase Grouter's scalability on even larger architectures, we performed pre-training on a 16B model for 10B tokens.
> > >
> > >
> > > | Method / Token | 1B | 2B | 3B | 4B | 5B | 6B | 7B | 8B | 9B | 10B |
> > > | --- | --- | --- | --- | --- | --- | --- | --- | --- | --- | --- |
> > > | ZL  | 4.423 | 3.550 | 3.210 | 3.102 | 3.030 | 2.945 | 2.907 | 2.826 | 2.817 | 2.806 |
> > > | Grouter | **3.902** | **2.872** | **2.421** | **2.224** | **2.067** | **1.976** | **1.903** | **1.818** | **1.793** | **1.762** |
> > >
> > >
> > > As illustrated, even when the model size scales to 16B, Grouter significantly accelerates the training process.
> > >
> > > ### The analysis of Grouter's effect for model with different scale
> > >
> > > To further analyze the impact of model size, we compare the final training loss (at 10B tokens) across different parameter scales:
> > >
> > > | Parameter | 350m  | 1B | 3B | 7B | 16B |
> > > | ---  | --- | --- | --- | --- | --- |
> > > | Loss w/o Grouter | 3.563  | 3.273 |3.170 | 2.995| 2.806|
> > > | Loss w Grouter | **3.439** | **2.900** | **2.306** | **2.065** | **1.762** |
> > > |Relative Loss Ratio| 96.52% | 88.60% | 72.74% | 68.94% | 62.79% |
> > >
> > >
> > > **Key Observation**: The data clearly demonstrates that Grouter's performance does not degrade as model size increases. On the contrary, the **acceleration benefits and loss reduction become even more pronounced at larger scales**, highlighting its exceptional scalability.
> > >
> > > We have provided a preliminary discussion of this phenomenon in Appendix K. We believe the mechanism behind why Grouter exhibits increasingly superior acceleration capabilities as model size scales is a compelling and significant direction for future research.
> > >
> > > ---
> > > We sincerely appreciate the reviewer’s thoughtful comments once again. We hope the above responses fully address the concerns raised.

---

### Official Review · Reviewer_MLWW · 2026-03-04

**Soundness:** 4
**Presentation:** 3
**Significance:** 4
**Originality:** 4
**Overall Recommendation:** 4
**Confidence:** 3

**Summary:**

The standard training process for MoE models involves training the router and the expert layers simultaneously. This means that the router is constantly changing how it routes tokens to the experts, which means experts have to train on constanly shifting partitions of tokens and makes it difficult for specialization. This work proposes Grouter, a pretrained general router for MoE models. The authors propose decoupling the training of the MoE router from the remaining network layers. The router is first trained through knowledge distillation using a large, fully converged source MoE model. The trained router is then frozen and used to train the remaining parts of the target MoE model. Evaluation results show that Grouter achieves the same validation loss as baseline models using only 23.3% of the training data, alongside up to a 33.5% increase in training throughput.

**Compliance With Llm Reviewing Policy:**

Affirmed.

**Final Justification:**

The rebuttal has addressed my primary concerns. I am keeping my positive score.

**Key Questions For Authors:**

- Can Grouter generalize to models that have more experts than the base model, or does it only work on smaller models?
- The preemptive routing strategy for Expert Parallelism requires extensive data preprocessing and caching. What is the specific overhead associated with this approach in terms of total processing time and additional memory or storage requirements?
- How effectively does a single Grouter instance generalize across a diverse range of target models, particularly those specializing in distinct tasks or disparate data distributions?

**Limitations:**

Includes an impact statement and a discussion of future work.

**Strengths And Weaknesses:**

Strength:
- This paper introduces Grouter, which challenges conventional approaches to training MoE models. The core idea is highly impactful: by decoupling the training of the router and the experts, Grouter ensures that experts can specialize on appropriate subsets of tokens. As a result, it reaches comparable validation loss much faster and achieves lower validation loss at the same training iteration.
- The paper also proposes preemptive routing for expert parallelism. Because the router is frozen for the remainder of training, the system can cluster experts and assign sequences to clusters in a way that minimizes overall communication, leading to improved throughput.
- Grouter is shared across all layers and can scale to different numbers of experts, allowing it to adapt flexibly to different model configurations.

Weaknesses

-  While the authors claim that Grouter's performance advantages expand with model scale , the evaluated models are quite small. How is the accuracy impact affected when moving to significantly larger architectures, such as Mixtral 8x7B or 8x22B? Also, how well does the distilled router generalize to target models that possess a significantly higher expert count than the original source model?

- There are two primary challenges I see with decoupling the router from expert training. While this method improves overall training efficiency, there are still advantages to co-optimizing these components. Specifically, different models often specialise in unique data subsets. This is particularly relevant for the smaller models that  seems to be the primary use case for Grouter. With Grouter, the routing structure is distilled from a larger, potentially more generalized MoE model. When transferring this router to a smaller model targeting a more specialized dataset, the lightweight "expert tuning" phase, may be insufficient to adapt to the new data distribution. This could result in suboptimal routing strategies compared to a joint optimization approach where the router and experts are trained together. This can also lead to significant load balancing issues, which is another challenge in efficient MoE training techniques

---

> ### Author Rebuttal · Authors · 2026-03-31
>
> We greatly appreciate your comments and feedback.
> ### 1. On Model Scaling and High Expert Count (Addressing Weakness 1 & Question 1)
>
> To handle cases where the target model has more experts than the source ($E_T > E_S$), we propose a "Fill-the-Gap" expansion strategy:
>
> - Identify Gaps: We identify "under-represented" tokens from a calibration set—those that the current $E_S$ experts fail to route with high confidence (low max-logits).
>
> - Assign New Experts: We perform K-Means clustering on the hidden states of these tokens. The resulting $E_{new}$ centroids are used to initialize the weights of the new experts, ensuring they specifically target previously neglected semantic regions.
>
> - Ensure Diversity: We project these new weights onto the null space of the existing experts and apply QR-orthogonalization. This ensures that the new experts are functionally distinct and non-redundant relative to the original ones.Why it works: Instead of random initialization, our method expands the routing space by strategically "filling the semantic gaps." This allows the Grouter to adapt to a larger expert pool ($E_T$) almost instantly, maintaining high routing quality and load balance from the start of training.
>
> We conduct the detailed experiment to verify this method and the effect of scaling model size. However, because of the word limit, please refer to the first point in our response to Reviewer mByH and second point to Reviewer Jir5 for the experiment.
>
> ### 2. On Data Distribution and Optimization Trade-offs (Addressing Weakness 2 & Question 3)
>
> Please refer to the third point of our response of mByH
>
> ### 3. On Overhead of Preemptive Routing (Addressing Question 2)
>
> 1. Storage Overhead of Preemptive RoutingTo store routing metadata, we use uint8 for Expert IDs and bf16 for routing scores. The storage requirement is approximately $E_{act} \times 3$ bytes per token (where $E_{act}$ is the number of active experts). For the Qwen3 configuration ($E_{act}=8$), this equates to 24 bytes per token.
>
>
> For a trillion-token dataset, this requires roughly 10-20 TB of storage. While non-trivial, this is well within the capacity of modern industrial clusters, where storage is typically measured in Petabytes (PB). Furthermore, this data does not need to be stored concurrently; pre-dispatching can be performed in streaming batches, allowing metadata for processed data to be deleted periodically to minimize the active storage footprint.
>
> 2. Computational Cost Analysis (FLOPs)
>
> We estimate the computational cost following the GPT-3 framework [1]. Let $T = b \times s$ (total tokens) and $P = 12lh^2$ (dense parameter estimation). The costs for forward and backward passes are estimated as $2TP$ and $4TP$, respectively. For MoE models, we apply a scaling coefficient $k = \frac{E_{act}}{E_{total}}$ to refine the estimate.
>
> Table R3: Detailed Breakdown of Computational Costs
> | Stage | Cost (TFLOPs) | % vs. Direct Pre-training |
> | :--- | :--- | :--- |
> | Grouter Distillation | 150,995 | 4.00% |
> | Expert Folding (Optional) | 981 | 0.03% |
> | Expert Tuning | 2,944 | 0.08% |
> | Preemptive Routing | 113,246 | 3.00% |
> | Pre-training with Grouter (6B tokens) | 452,983 | 12.00% |
> | Total Grouter Pipeline | 721,149 | 19.11% |
> | Direct Pre-training (50B tokens) | 3,774,873 | 100.00% |
>
> 3. Key Takeaways on Efficiency
> - 81% Computation Reduction: As shown in Table R3, the end-to-end Grouter pipeline requires only 19% of the FLOPs compared to the baseline to reach the same loss, representing an 81% reduction in total computation.
> - Negligible Overhead: The additional steps (Distillation, Folding, Tuning, and Preemptive Routing) combined account for only $\sim$7% of the original training budget.Resource Optimization: GPU Idle Time: Preemptive routing can be performed during GPU idle periods, further reducing the perceived cost.
>     * CPU-Bound Tasks: Operations such as Expert Grouping and Sample Placement Optimization are purely CPU-based scientific computations. Since CPU resources in GPU clusters are often underutilized, these can be offloaded and executed in parallel with pre-dispatching without bottlenecking the training process.

---

> > ### Author Rebuttal · Reviewer_MLWW · 2026-04-01
> >
> > Thank you for the thorough response. My primary concerns are fully addressed, hence I would keep my positive score.

---

> > > ### Author Response · Authors · 2026-04-02
> > >
> > > Thank you very much for your acknowledgment and for the time you dedicated to reviewing our rebuttal. We are encouraged to hear that our responses and the additional clarifications on model scaling, data distribution, and preemptive routing overhead have fully addressed your primary concerns. we are also grateful that the additional experiments and analysis are considered helpful for our next revision.

---

### Official Review · Reviewer_Jir5 · 2026-03-07

**Soundness:** 2
**Presentation:** 4
**Significance:** 2
**Originality:** 3
**Overall Recommendation:** 5
**Confidence:** 4

**Summary:**

This paper proposes GROUTER, a decoupled routing framework for MoE training that first distills a fixed router from a well-trained source MoE model and then uses it to guide the training of a new target MoE. The main motivation is that jointly learning routing and expert representations causes optimization interference and unstable convergence. To address this, the paper extracts a routing prior from a converged source model, and further introduces Expert Folding and Expert Tuning to adapt the router to target models with different expert configurations. Empirically, the method shows improved convergence speed, lower validation loss, better training stability, and additional throughput gains under suitable system settings.

**Compliance With Llm Reviewing Policy:**

Affirmed.

**Final Justification:**

The rebuttal has addressed my concerns, and I keep the score.

**Key Questions For Authors:**

1. If a well-trained source MoE model is already available for training the Grouter, why not directly continue fine-tuning that MoE model instead of training a new model based on the distilled routing structure? (This is the main concern, and I am looking forward to your response.)
2. Since the Grouter is an external routing module for the MoE model and appears effective in Figure 5(b), can it also be applied to the original MoE model (e.g., Qwen3 in the paper) to accelerate inference?
3. “Tiny-Qwen3” or “Mini-Qwen3” in Section 4.1?

**Strengths And Weaknesses:**

Strengthen:

1. GROUTER significantly reduces the validation loss throughout training, thereby improving training stability and robustness.
2. Figures 1(a) and 1(b) are well presented, and the observed results are reasonable and consistent with the paper’s claims.
3. The method is lightweight on the target-model side at deployment time, since the distilled router is relatively small compared with the full MoE model. Moreover, the paper demonstrates consistent gains across several target architectures and scales.

Weaknesses:

1.  The method requires an additional well-trained source MoE model to train the Grouter.
2. The method does not handle structural mismatches between the source and target MoE models very well. Although the paper discusses the case where the target MoE has fewer experts than the source MoE, it does not consider the opposite case, nor does it address cases where the target model has more or fewer layers.
3. The cost-benefit tradeoff is not fully analyzed. Although the paper shows faster convergence for the target model, it remains unclear whether the overall pipeline is still advantageous after accounting for the cost of training the source MoE and distilling the Grouter.

---

> ### Author Rebuttal · Authors · 2026-03-31
>
> We greatly appreciate your comments and feedback.
> ### 1. On the Motivation of Training from Scratch vs. Fine-tuning (Addressing Question 1)
> We thank the reviewer for this insightful point. While fine-tuning is an established practice when resources and architectures are perfectly aligned, GROUTER addresses critical scenarios where direct fine-tuning is either infeasible or suboptimal. Our framework provides unique value in the following contexts:
> - **Extreme Inference Efficiency & Edge Deployment (Model Downsizing)**:
>      * Fine-tuning can adapt a model’s capabilities but cannot alter its underlying parameter scale or computational complexity. A massive source MoE (e.g., 100B+) remains undeployable on memory-constrained devices like smartphones or PCs, regardless of how much it is fine-tuned.
> - **Breaking the Fine-tuning Ceiling & Avoiding Pre-training Bias**:
>     * The Limitation of Fine-tuning: Directly fine-tuning a massive model on domain-specific data often suffers from "pre-training inertia," where the model's original weights restrict its ability to fully re-specialize to a new distribution. Furthermore, many parameters in a giant model may be redundant for specialized tasks, leading to wasted computation.
> ### 2. On Structural Robustness and Layer Mismatch (Addressing Weakness 2)
>
> There appears to be a misunderstanding regarding our architecture. As specified in Section 3.1.2, Grouter does not require a layer-by-layer mapping between models. Instead, we use a single, shared Grouter to govern the routing decisions for all layers of the Target MoE.
>
> To address scenarios where the target model has more experts than the source ($E_T > E_S$), we developed the Expert Expanding strategy. This was achieved by identifying the null space (complementary space) of existing expert weights and initializing new experts to cover previously under-represented semantic regions. (The detail in the first point of our response to reviewer MLWW due to word limit)
>
> Specifically, we expanded the expert count of a Grouter distilled from Qwen3-30B-A3Bto 256 experts and below is the result
> | Method / Token | 1B | 2B | 3B | 4B | 5B | 6B | 7B | 8B | 9B | 10B |
> | --- | --- | --- | --- | --- | --- | --- | --- | --- | --- | --- |
> | Baseline (7B, 256 experts) | 6.332 | 4.626 | 3.729 | 3.459 | 3.306 | 3.199 | 3.134 | 3.076 | 3.032 | 2.995 |
> | Grouter + Expanding (7B, 256 experts) | 6.055 | 4.260 | 3.364 | 2.864 | 2.595 | 2.398 | 2.273 | 2.189 | 2.101 | 2.065 |
>
> The results clearly demonstrate that Grouter delivers a substantial training acceleration.
> ### 3. On the Overall Pipeline Cost and Efficiency (Addressing Weakness 1 & Weakness 3)
> Grouter is designed to extract structural priors from high-quality open-source models to accelerate pre-training. Similar to conventional distillation, the acquisition cost of the source model is excluded from Grouter’s training overhead.
>
> Below is an end-to-end breakdown demonstrating the computational savings of Grouter-based pre-training.
>
>
> To demonstrate the efficiency of GROUTER, we provide a detailed FLOPs breakdown based on the GPT-3 estimation framework. We compare a standard 50B-token pre-training of Mini-Qwen3 against a GROUTER-assisted run (achieving the same loss in only 6B tokens).
> Distillation ($C_{dist}$): Requires forward/backward passes on Grouter and a forward pass on the source model's first layer only ($n=1$). For 2.4B tokens:
> $$C_{distillation} \approx 1.5 \times 10^5 \text{TFLOPs}$$
> Expert Folding & Fine-tuning ($C_{EF}$): Applied once using 52M tokens:
> $$C_{EF} = C_{Fold} + C_{Fine} \approx 3.9 \times 10^3 \text{ TFLOPs}$$
> Baseline (50B tokens): Standard pre-training requires:
> $$ C_{train} \approx 3.77 \times 10 ^6 \text{TFLOPs}$$
> ROUTER-Assisted (6B tokens): Total cost including Grouter inference:$$C_{G\_total} = C_{train} + C_{G\_inf} \approx 5.66 \times 10^5 \text{ TFLOPs}$$
> Therefore, the full Grouter pipeline requires approximately 721,151 TFLOPs, compared with 3,774,873 TFLOPs without Grouter — representing only 19% of the original computational cost.
> We further decompose the training process and observe that the additional operations introduced by Grouter do not account for a major portion of the total training cost.
> | Stage | Cost(TF) | Percentage versus Pretraining Directly|
> | :--- | :--- | :--- |
> | Distillation | 150995 | 4.00% |
> | Expert Folding | 981 | 0.03% |
> | Expert Tuning | 2944 | 0.08% |
> | PreTraining with Grouter| 566231 | 15.00% |
> | PreTraining Directly | 3774873 | 100.00% |
> ### 4.Others (Addressing Q2 & 3)
> It is important to clarify that Grouter is not designed to accelerate the training of the Source Model itself. The routing logic and expert weights in the original Source MoE are already deeply coupled during its own pre-training. So Grouter can't work.
> We used Tiny-Qwen3 in Section 4.1. We apologize for not clearly indicating this in the experimental setup, and we will supplement this information in the experimental description in the revised version.

---

> > ### Author Rebuttal · Reviewer_Jir5 · 2026-04-01
> >
> > Thanks for your rebuttal and clarification. My concerns have been addressed, and I have raised my score accordingly.

---

> > > ### Author Response · Authors · 2026-04-02
> > >
> > > Thank you for the update and for recognizing that our clarifications on the motivation for training from scratch and the end-to-end efficiency analysis have fully addressed your concerns; we are also grateful that the additional experiments on structural robustness and the Expert Expanding strategy are considered helpful for our next revision. We sincerely appreciate your support and the reassessment of our work.

---

### Official Review · Reviewer_mByH · 2026-03-08

**Soundness:** 3
**Presentation:** 3
**Significance:** 4
**Originality:** 4
**Overall Recommendation:** 5
**Confidence:** 4

**Summary:**

This paper addresses the inefficiencies with MoE training, particularly the challenges faced when learning routing policy with representation learning. The authors propose General Router (GRouter), which learns the structural knowledge via distillation from a fully trained MoE model. Grouter then acts as a plug-and-play replacement for training different MoE models, significantly accelerating the training process since the focus is only on learning the representation. The authors propose expert folding and expert tuning to support transferability of Grouter to different downstream MoE configurations than the teacher MoE. The decoupling also enables several training optimizations, which effectively reduce the training time and enhance throughput. Several experiments and ablations confirm the benefits of Grouter.

**Compliance With Llm Reviewing Policy:**

Affirmed.

**Final Justification:**

The rebuttal convincingly addressed my concerns, and I will therefore maintain my positive score.

This is a strong paper with a highly original contribution that has the potential to influence how future MoE models are trained. If needed, I would be happy to champion its acceptance.

**Key Questions For Authors:**

1. It appears that expert folding is applicable for scenarios where $E_T < E_S$. Can the design also support $E_T > E_S$? Is this desirable?
2. The distillation dataset seems to be the same as the pre-training dataset. Would it be possible to use a different dataset for distillation? It would be interesting to see how prominent expert tuning is in such a scenario.
3. Could the authors comment on whether picking a different layer for distillation (maybe not too far from the first layer) would provide performance benefits? Was such a behaviour observed in the experiments?

**Limitations:**

Please see my weaknesses description above.

**Strengths And Weaknesses:**

## Strengths

1. This is a well written paper with a clear motivation.
2. The idea of decoupling of routing from representation learning is very interesting and the authors do a commendable job in actually making it work with several novel amendments. The benefits of such an approach are clearly demonstrated via extensive experiments, demonstrating not only 4x faster training but also lower final loss.
3. The architectural design of Grouter is thoroughly explored, with several ablations reported in Appendix G.
4. The design of Expert Folding and Expert Tuning is also fairly neat. Indeed, when the target distribution is different from the source distribution from which Grouter is distilled, such a fine-tuning makes a lot of sense. Further, the authors clearly empirically demonstrate the benefits of expert tuning in improving load balancing (Figure 4b) as well as the benefits of both components in achieving transferability to different model configurations (Appendix K).
5. The decoupled router is also cleverly used to enable further runtime acceleration during training by caching routing decisions for each input and including them as a part of the pre-processed dataset. This neat design allows for data placement optimization where samples are placed to be close to their EP, thereby reducing communication costs.
6. The experimental section is thorough, covering several ablations, performance on downstream tasks, and efficiency gains.

Overall, this is a very strong paper with a neat idea and clear execution of experiments. I would recommend acceptance of this paper. Some areas can still be improved or further discussed which I will list below.

## Weaknesses

1. The proposed design does not support variable routing across layers and might not perform well when such variability is desirable.
2. The size of the teacher model from which Grouter is distilled (30B) is one order bigger than considered pre-training model configurations (biggest 3B). It would be interesting to see whether the benefits still hold for pre-training bigger models.
3. Many of the experimental details are deferred to the Appendix, which makes it difficult to follow the results. The reported baselines are not described or cited in the main experimental text (though they are briefly mentioned in the Introduction). I would highly recommend including parts of Appendix E and F in the main paper.
4. All the experiments focus on only one dataset (C4). Inclusion of another dataset will strengthen the generalizability of the results.

---

> ### Author Rebuttal · Authors · 2026-03-31
>
> We greatly appreciate your comments and feedback.
>
> ### 1. On Model Scaling and Expert Compatibility (Addressing Weakness 2 & Question 1)
>
> We denote the sizes of the Target MoE and Source MoE as $P_T$ and $P_S$, respectively. The generalizability of Grouter across model scales encompasses two distinct scenarios and we conduct experiments under both scenarios:
> **(i) Upward Scaling ($P_T > P_S$), where a Grouter distilled from a smaller model accelerates the training of a larger target model;**
>
> we first pre-trained a 550M Tiny-Qwen3 model on 30B tokens to serve as the source. We then distilled the routing prior from its $0^{th}$ layer to construct the Grouter. Subsequently, this Grouter was incorporated to guide the pre-training of a 1B Mini-DS-V2-Lite target model.:
>
> | Method / Tokens | 2B | 4B | 6B | 8B | 10B | 12B | 14B | 16B | 18B | 20B |
> | ------ | --- | --- | --- | --- | --- | --- | --- | --- | --- | --- |
> | Z Loss + Aux Loss | 5.476828 | 3.970089 | 3.539572 | 3.377311 | 3.273886 | 3.230883 | 3.184982 | 3.137000 | 3.095379 | 3.089645 |
> | Grouter\_{Self} | 5.149749 | 3.951091 | 3.568707 | 3.398479 | 3.280460 | 3.241685 | 3.217345 | 3.167768 | 3.103344 | 3.077095 |
>
>
> As illustrated, even in the $P_T = 2P_S$ scenario (Upward Scaling), our results show that Grouter-based training achieves a validation loss comparable to the baseline at the same number of steps. Furthermore, by incorporating preemptive routing, we significantly enhance the training throughput, thereby accelerating the overall training process.
>
> **(ii) Downward Scaling ($P_S > P_T$), where a Grouter derived from a larger source model guides the training of a relatively smaller target model(differ by less than an order of magnitude).**
>
> Due to space limitations, please refer to the second point in our response to Reviewer Jir5.
>
> ### 2. On Routing Variability and Structural Design (Addressing Weakness 1 & Question 3)
>
>
>
> We carefully considered the implementation of layer-wise routing during the design phase but ultimately opted for a shared Grouter for the following reasons:
>
> - **Redundancy in Layer-wise Routing**: Recent studies have indicated significant redundancy in routing decisions across different MoE layers. Consequently, the marginal gain in representational capacity from independent routing per layer is often outweighed by the increased complexity.
>
> - **System-level Communication Bottlenecks**: Layer-wise routing poses a substantial burden on communication optimization.
>
>
> - **Empirical Success of the Unified Grouter**: Our experimental results demonstrate that a unified Grouter achieves a 4.28$\times$ improvement in data efficiency and substantially elevates the convergence ceiling of the MoE model.
>
>
> While we chose the first layer for the aforementioned reasons, other layers can also serve as distillation sources. To provide a comprehensive view, we present the training results on Tiny-Qwen3 using Grouter instances distilled from different layers
>
>
> | Distilled Layer / Token | 1B | 2B | 3B | 4B | 5B | 6B | 7B | 8B | 9B | 10B |
> | --- | --- | --- | --- | --- | --- | --- | --- | --- | --- | --- |
> | Layer 0 | 5.337 | 4.158 | 3.696 | 3.423 | 3.258 | 3.166 | 3.084 | 3.047 | 3.041 | 2.999 |
> | Layer 3| 5.288 | 4.151 | 3.762 | 3.541 | 3.406 | 3.336 | 3.271 | 3.246 | 3.246 | 3.210 |
> | Layer 7 | 5.341 | 4.182 | 3.802 | 3.595 | 3.462 | 3.396 | 3.337 | 3.317 | 3.318 | 3.283 |
>
> It can be seen from the experimental results that the Grouter distilled from earlier layers yields better performance.
>
>
> ### 3. On Generalizability and Adaptation to Data Distribution (Addressing Weakness 4 & Question 2)
>
> We conducted experiments using SlimPajama  and OpenWebMath for both Expert Tuning and subsequent pre-training. This allows us to assess Grouter's adaptability when transitioning from general to highly specialized domains. The results are summarized below:
>
>
> | Dataset | Method | 1B | 2B | 3B | 4B | 5B | 6B | 7B | 8B | 9B | 10B |
> | --- | --- | --- | --- | --- | --- | --- | --- | --- | --- | --- | --- |
> | OpenWebMath | ZL | 5.006 | 3.356 | 3.006 | 2.825 | 2.746 | 2.684 | 2.645 | 2.650 | 2.607 | 2.586 |
> | OpenWebMath | Grouter | **4.505** | **3.213** | **2.865** | **2.628** | **2.510** | **2.414** | **2.360** | **2.348** | **2.305** | **2.274** |
> | SlimPajama | ZL | 5.788 | 4.236 | 3.779 | 3.595 | 3.453 | 3.401 | 3.373 | 3.348 | 3.324 | 3.329 |
> | SlimPajama | Grouter | **5.384** | **4.058** | **3.591** | **3.315** | **3.127** | **3.023** | **2.964** | **2.924** | **2.886** | **2.872** |
>
>
>
> To further demonstrate the effect of Expert Tuning, we tested the load balancing level of Grouter before and after Expert Tuning:
>
>
> | Dataset | MaxVio (Before EF) | MaxVio (After EF) |
> | --- | --- | --- |
> | OpenWebMath | 2.7953 | 0.6490 |
> | SlimPajama | 2.5271 | 0.5299 |

---

> > ### Author Rebuttal · Reviewer_mByH · 2026-04-03
> >
> > Thank you for the detailed responses. I appreciate the experiments on the expert expansion strategy included in the response to reviewer Jir5. Most of my concerns have been addressed, and thus would like to maintain my positive score.

---

> > > ### Author Response · Authors · 2026-04-04
> > >
> > > Thank you for the update and for your meticulous review of our collective responses; we are particularly encouraged that the "Expert Expansion" strategy detailed in our response to Reviewer Jir5 resonated with your initial concerns regarding structural compatibility. We are also grateful that our additional experiments on model scaling and cross-dataset generalizability have fully resolved the remaining points, and we look forward to incorporating these multi-faceted analyses into the next revision.

---

### Decision · Program_Chairs · 2026-04-30

**Decision:**

Accept (regular)

**Comment:**

This paper proposes GROUTER, a decoupled routing framework for MoE training that distills routing structure from a converged source MoE and reuses it to guide target-model training. Reviewers generally agreed that this is a well-motivated and interesting idea, with substantial practical value for improving MoE training efficiency and stability.

The main strengths are the originality of the decoupled routing perspective, the strong empirical gains in data efficiency and throughput, and the additional mechanisms for transferability and systems optimization. Most reviewers found the work technically solid and potentially impactful.

The main concerns were about scaling evidence, support for target models with more experts than the source, end-to-end overhead, and presentation clarity. Based on the rebuttal and discussion, I believe these concerns were addressed sufficiently: the authors provided additional scaling results beyond the original 3B range, clarified their expert expansion strategy, and included a more complete efficiency analysis. While some presentation issues remain, these appear revisable and do not undermine the core contribution.

Overall, I find this to be a strong and useful contribution with broad reviewer support. For the final version, the authors should better integrate the added scaling and overhead analyses into the main paper and further improve the clarity of the pipeline description.